# A differential emissivity imaging technique for measuring hydrometeor mass and type

Dhiraj K. Singh[1], Spencer Donovan[1], Eric R. Pardyjak[1], and Timothy J. Garrett[2]

[1]Department of Mechanical Engineering, University of Utah, Salt Lake City, UT, USA
[2]Department of Atmospheric Sciences, University of Utah, Salt Lake City, UT, USA

**Correspondence:** Eric R. Pardyjak
(pardyjak@eng.utah.edu)

**Abstract.** The Differential Emissivity Imaging Disdrometer (DEID) is a new evaporation-based optical and thermal instrument designed to measure the mass, size, density, and type of individual hydrometeors and their bulk properties. Hydrometeor spatial dimensions are measured on a heated metal plate using an infrared camera by exploiting the much higher thermal emissivity of water compared with metal. As a melted hydrometeor evaporates, its mass can be directly related to the loss of heat from the hotplate assuming energy conservation across the hydrometeor. The heat-loss required to evaporate a hydrometeor is found to be independent of environmental conditions including ambient wind velocity, moisture level, and temperature. The difference in heat loss for snow versus rain for a given mass offers a method for discriminating precipitation phase. The DEID measures hydrometeors at sampling frequencies up to 1 Hz with masses and effective diameters greater than 1 μg and 200 μm, respectively, determined by the size of the hotplate and the thermal camera specifications. Measurable snow water equivalent (SWE) precipitation rates range from 0.001 to 200 mm h$^{-1}$, as validated against a standard weighing bucket. Preliminary field-experiment measurements of snow and rain from the winters of 2019 and 2020 provided continuous automated measurements of precipitation rate, snow density, and visibility. Measured hydrometeor size distributions agree well with canonical results described in the literature.

## 1 Introduction

Accurate measurements of the mass, density, shape, size and precipitation rate of hydrometeors are critical for scientific, industrial, and commercial applications, as well as weather prediction. Falling hydrometeors play an essential role in daily human activity, with impacts ranging from the hydrological cycle (Stendel and Arpe, 1997) to transportation (Campbell and Langevin, 1995; Theofilatos and Yannis, 2014). Ground-based weighing gauges can provide measurements of precipitation rate (Golubev, 1985a, b; Goodison et al., 1989; Yang et al., 1998; Brock and Richardson, 2001) but often require anti-freeze additives with a glycol-based solution and oil skim overlays to prevent evaporation of water from the solution, or require manual emptying during a storm (Finklin, 1988). Optical gauges (Deshler, 1988; Loffler-Mang and Joss, 2000; Gultepe and Milbrandt, 2010) have the advantage of measuring the size of hydrometeors in free fall but tend to work better for rain than for snow due to the wide variation of particle density (Pomeroy and Gray, 1995; Judson and Doesken, 2000), which introduces large uncertainties in the measurement of snow water equivalent (SWE) and snow precipitation rate (Brandes et al., 2007; Lempio

et al., 2007). Other instruments used for quantifying precipitation rate, fall speed, size distribution, and visibility include the hotplate precipitation gauge (Rasmussen et al., 2011), the Multi Angle Snowflake Camera (Garrett et al., 2012; Notaroš et al., 2016; Fitch et al., 2021) 2DVD (Kruger and Krajewski, 2002; Randeu et al., 2013), and the PARSIVEL (Battaglia et al., 2010; Friedrich et al., 2013; Loeb et al., 2021). However, none of these instruments measure the mass and density of individual hydrometeors, which is essential for accurate prediction of fall speed as well as avalanche safety issues (Brun et al., 1989). Accurate measurement of precipitation rate is more difficult for solid hydrometeors due to numerous factors, including losses from evaporation, wind, and wetting (Sevruk and Klemm, 1989; Yang et al., 2005; Rasmussen and Coauthors, 2012) and significant changes in precipitation rate due to high wind speeds during storms (Yang et al., 1999). To overcome the effect of wind on precipitation measurement, various wind shields have been used around gauges (Goodison et al., 1998; Yang, 2014). Another critical parameter that affects precipitation rate measurement is the *catch efficiency* of snow, which depends on wind speed, snowflake density, and type (Colli et al., 2015). Many instruments have a minimum threshold to measure precipitation rate and this creates difficulties particularly in the cold northern latitudes, where the snow fall intensities are relatively low.

Understanding the size distribution of hydrometeors is an important consideration related to the physics of precipitation. Hydrometeor size has been measured using optical techniques (Knollenberg, 1970) and fall speed (Locatelli and Hobbs, 1974). Barthazy et al. (2004) measured the size and fall speeds of hydrometeors greater than 1 mm in diameter using optics-based instruments.

While the instruments described above measure many key precipitation variables, there is not a single device capable of measuring individual hydrometeor mass and density. Here, we present a new ground-based instrument, the Differential Emissivity Imaging Disdrometer (DEID), for the measurement of mass, shape, density, and size of individual hydrometeors, as well as integrated quantities such as precipitation rate and visibility. The DEID measures particle-by-particle physical properties of hydrometeors with high accuracy and is insensitive to environmental conditions (i.e., wind speed, temperature, and humidity). The DEID is designed to accurately measure individual hydrometeors with diameters greater than 0.2 mm and masses greater than 1μg, which includes approximately all sizes and types of falling hydrometeors. Hydrometeor size distributions can be determined for rain by using the effective spherical diameters inferred from the mass measurement and density of water. A heat flux parameterization is used to discriminate the type of precipitation (rain, snow, and mixture) and the ratio of actual area to circumscribed area over the maximum size of a snowflake is used to discriminate the type of snow.

This paper is organized as follows. In section 2, the theory behind the DEID's measurement methodology is presented. The experimental set-up and data processing are discussed in section 3. In section 4, the basic DEID measurements are validated using laboratory experiments. Density, SWE, and size are discussed in section 5 and the size distribution of rain droplets and snowflakes are described in section 6. A summary and several conclusions are given in section 7.

## 2 Background theory: The DEID measurement methodology

### 2.1 Hydrometeor mass measurement

The DEID consists of a temperature-controlled hotplate with a low-emissivity ($\epsilon$) top surface and a thermal camera. Figure 1 includes a schematic of the basic DEID set-up and a photograph of the DEID deployed in a field experiment. The grayscale thermal images of the hotplate without hydrometeors look dark due to its low emissivity and hence low brightness temperature. When water droplets are applied to the hotplate, they appear bright due to their high $\epsilon$ and high temperature. This creates excellent contrast that enables the measurement of the hydrometeor's size and area by counting pixels. The working principle of the DEID is based on conservation of thermal energy for a control volume taken around a hydrometeor (see Fig. 2). When a hydrometeor falls on the hotplate ($\approx 100°\text{C}$), it evaporates and its mass is directly related to the loss of heat from the hotplate. We assume that heat loss from the hotplate is conductive and one dimensional. And the heat gain by the hydrometeor is equivalent to heat loss from the hotplate sufficient for evaporation. The conductive heat flow from the hotplate to liquid or solid hydrometeors is a function of thermal conductivity of the plate ($k_{AL}$), the thickness of the plate ($d_{AL}$), the temperature difference between the bottom ($T_b$) and top of the plate ($T_p$), the plan area of the hydrometeor (cross sectional area perpendicular to the heat flow) on the hotplate at a time $t$ ($A(t)$) and evaporation time $\Delta t$.

The energy balance across a hydrometeor that falls onto the hotplate may be written as

*Heat gain by hydrometeor = Heat loss from hotplate.* (1)

Considering a control volume wrapped around a droplet as shown in Fig. 2, the droplet energy balance includes: energy storage, evaporation, conduction, convection and radiation and may be written as

$$c\Delta T \int dm + L_v \int dm = \int_0^{\Delta t} \frac{k_{AL}}{d_{AL}} A(t)(T_b(t) - T_p(t))dt$$
$$- \int_0^{\Delta t} h_c A \Delta T dt - \epsilon_w \sigma b \int_0^{\Delta t} A(t)(T_w^4(t) - T_{air}^4)dt, \tag{2}$$

where $T_w$ is the temperature of the water droplet, $\Delta T$ is the temperature difference between the initial and final temperature of the water droplet, $c$ is the specific heat capacity of water, $L_v$ is the latent heat of vaporization of water, $dt$ (approximated as $\Delta t$) is the time required to evaporate the water droplet, $T_{air}$ is surrounding air temperature , $h_c$ is the convective heat transfer coefficient, $\epsilon_w$ is emissivity of water, $\sigma$ the Stefan-Boltzmann constant, $b$ is the radiation view factor of 0.66 (Feingold, 1966), and $m$ the hydrometeor mass. The mechanics of the heat gain and loss by a hydrometeor is shown in Fig. 2a and the heat flow through plate and hydrometeor in Fig. 2b.

The cross-sectional area of the hydrometeor normal to the fall velocity direction (plan view) is measured with a thermal camera after it lands on the hotplate by taking advantage of the differential emissivity between the metal plate and the hydrometeor. Hydrometeors have a near unity emissivity whereas the emissivity of aluminum is near zero, so hydrometeors appear as

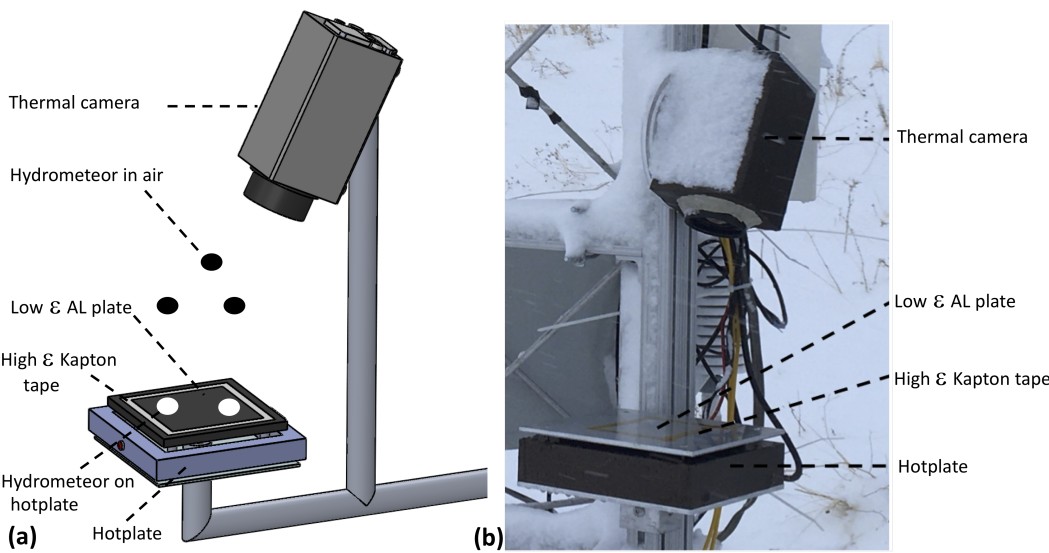

**Figure 1.** (a) Schematic of the DEID. The top surface of a roughened heated aluminum plate imaged by a thermal camera is dark due to its low infrared emissivity. Hydrometeors with a high emissivity that reach a high temperature on the heated plate show as bright spots from which the hydrometeor's size and area by counting pixels. (b) Photograph of the DEID after deployed in field experiments.

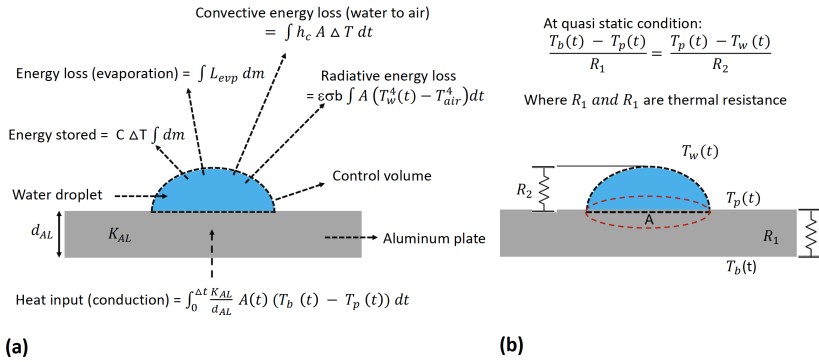

**Figure 2.** Schematic of energy conservation of control volume across hydrometeor. (a) Distribution of heat gain and loss from the hydrometeor. (b) Schematic of heat flow between the series combination the hotplate and water droplet at quasi static condition.

bright spots superimposed on a black background. In the case of snow, the particle size in air and after melting on the hotplate is quite similar, but can differ in the case of large rain droplets greater than 2 mm across. These considerations do not affect the calculation of mass through Eq. 2.

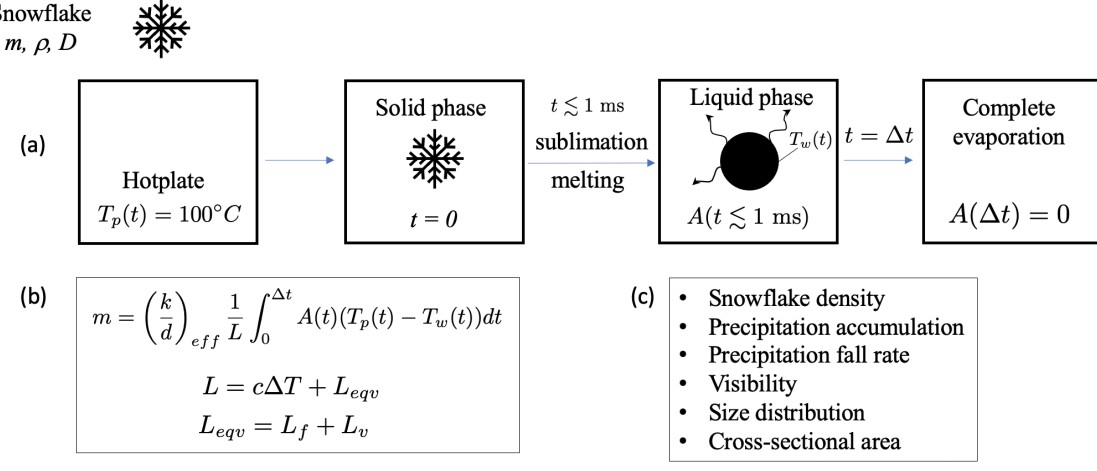

**Figure 3.** (a) Schematic illustrating the process of a snowflake falling onto the hotplate, melting, and evaporating. (b) DEID algorithm for measuring hydrometeor mass. (c) Output products deduced from the DEID measurements. $c$ is specific heat capacity of water, $\Delta T$ is temperature difference between initial and final water droplet temperature, $L_f$ is latent heat of fusion (e.g. sublimation),$L_v$ is latent heat of vaporization and $L_{eqv}$ is total latent heat of vaporization and fusion.

The temperature of the hotplate ($T_p$) is maintained at a temperature below the Leidenfrost temperature ($\approx 120°C$) so that heat transfer to the hydrometeors is maximized (Bergman et al., 2011). A schematic of the algorithm used to calculate individual hydrometeor properties from the heat transfer physics is shown in Fig. 3. The contribution of convective and radiative heat loss during evaporation is very small compared to that from conductive heat loss. Assuming a typical value for the coefficient of convection in air based on the wind speed ($h_c = 10$ J m$^{-2}$K$^{-1}$s$^{-1}$), convective heat loss is $\approx 1\%$ and radiative heat loss is $\approx$ 1% of the total heat required to evaporate the given mass as described in the appendix.

Assuming convective and radiation losses are negligible, Eq. 2 can be re-written as

$$\text{Heat gain by hydrometeor} \approx \text{conductive loss from hotplate.} \tag{3}$$

$$c\Delta T \int dm + L_v \int dm \approx \int_0^{\Delta t} (k_{AL}/d_{AL})A(t)(T_b(t) - T_p(t))dt. \tag{4}$$

## 2.2 Statistics of individual hydrometeors

The equivalent circular diameter of a particle on the plate, $D_{eff}$, after impact and after melting is determined from the particle area through $A(t \approx 0) = (\pi/4)D_{eff}^2$, where $t \approx 0$ corresponds to the time when the thermal camera detects a bright spot on the plate associated with a hydrometeor. Typically there is a few millisecond lag between the actual impact and detection, as

verified by recording the processes at 240 Hz. $D_{eff}$ is nearly preserved after melting. This was verified by slowing down the melting process by reducing the hotplate temperature (40°C) and recording the processes at high frequency (120 fps). The size of approximately 2000 snowflakes were measured before and after melting. We found that the average change in $D_{eff}$ was 5%.

The maximum effective diameter $D_{max}$ is defined as the maximum dimension of the particle in the thermal camera two-dimensional plane. We also describe the first direct measurements of a melted diameter, $D_{mel}$ defined by the measured hydrometeor mass and the density of water (i.e., $(\pi/6)D_{mel}^3 = m/\rho_w$). Here, *particle complexity* is defined as the ratio of the area of the smallest ellipse completely containing the particle cross-section to the actual cross-sectional area of the hydrometeor measured on the hotplate. That is, *Complexity* $= ((\pi/4)D_{max}D_{min})/A(t \approx 0)$, where $D_{min}$ is the maximum dimension of the particle normal to the $D_{max}$. The complexity is always greater than or equal to unity, which corresponds to a circular shape. All the defined parameters are illustrated in Figure 7.

## 2.3 Measurement of SWE rate and accumulation

The instantaneous snow-water-equivalent (SWE) accumulation rate ($\dot{SWE}$) and the time-integrated SWE accumulation can be estimated on a frame-by-frame basis using the DEID. From the total mass of water deposited onto the hotplate in each frame, the SWE rate for a given time interval may be written as

$$\dot{SWE} = c_1 \frac{\Delta m \cdot fps}{\rho_w A_{hp}}, \tag{5}$$

where $c_1$ is conversion factor from m s$^{-1}$ to mm hr$^{-1}$ ($3.6 \times 10^6$ mm h$^{-1}$ m$^{-1}$ s), $fps$ is the image sampling rate in frames per second, $\Delta m$ (kg) is the total hydrometeor mass that falls on the hotplate in each recorded frame, $\rho_w$ (kg m$^{-3}$) is the bulk density of water and $A_{hp}$ (m$^2$) is a rectangular sampling area on the hotplate that captures many hydrometeors. To obtain the accumulated SWE, the rate is multiplied by the time interval between samples ($1/fps$) and then summed.

In addition to this frame-by-frame method, SWE and $\dot{SWE}$ can be estimated using a particle-by-particle method. In this case, $\Delta m$ in Eq. 5 is the total hydrometeor mass that falls on the hotplate over a given time interval $\Delta t$ in Eq. 4, which is sum of all individual hydrometeors that have completed the normal cycle of evaporation.

## 2.4 Measurement of individual snowflake density and snow precipitation rate

The density of individual snowflakes is given by $\rho_s = m/V$, where $m$ (kg) and $V$ (m$^{-3}$) are the mass and volume of an individual snowflake, respectively. The volume $V$ can be estimated by assuming a spherical particle of equivalent circular diameter $D_{eff}$ such that $V = (\pi/6)D_{eff}^3$. The density measurement of a snow layer after accumulation on the surface depends on many parameters that effect settling such as the overlying snow mass, surface properties and local weather parameters. However, an average density ($\overline{\rho}_s$) prior to settling over a given period can be calculated from DEID data using the ratio of the total mass to total volume in a given time interval, namely

$$\overline{\rho}_s = \frac{\sum_{i=1}^{N} m_i}{\sum_{i=1}^{N} m_i/\rho_{s,i}}, \tag{6}$$

where, $m_i$ (kg) is the mass of $i^{th}$ snowflake, $\rho_{s,i}$ (kg m$^{-3}$) the density of the $i^{th}$ snowflake and $N$ is the total number of snowflakes on the plate during the given time frame. From the average density of the snowflakes in each frame, the snow precipitation rate or precipitation intensity is:

$$PI_{snow} = c_1 \frac{\Delta m \cdot fps}{\overline{\rho}_s A_{hp}}. \tag{7}$$

Total snow accumulation is then the precipitation rate multiplied by the time interval between samples ($1/fps$) and then summed.

## 2.5 Measurement of visibility

Visibility can be estimated using the Koschmieder relation (Gultepe et al., 2009; Rasmussen et al., 1999). Specifically, the visibility (in cm) is calculated

$$Vis = \frac{C}{\beta_{ext}} \tag{8}$$

where $C = -\ln(0.05) = 2.996$, $\beta_{ext}$ is the path-averaged extinction coefficient of snow particles per unit volume (cm$^2$ cm$^{-3}$). The extinction coefficient per unit volume is define as

$$\beta_{ext} = \frac{\sum_{i=1}^{N} Q_{ext,i}(D_{eff},\lambda) A_i)}{V_a}, \tag{9}$$

where $N$ is total number of snowflakes that have fallen on the hotplate during time interval $\delta t$, $A_i$ (cm$^2$) is the area of the
145 $i^{th}$ snowflake and $V_a$ (cm$^3$) is the total sample volume of air in period $\delta t$, computed as $V_a = A_{hp} v_T \delta t$, where $v_T$ (cm s$^{-1}$) is the average snowflake terminal fall speed as described below. $Q_{ext,i}(D_{eff},\lambda)$ relates the physical cross-sectional area of snowflakes to the scattering cross-sectional area for visible wavelengths, which is $\approx 2$ for particles with sizes greater than 4 $\mu m$ (Gultepe et al., 2009). After substituting the equations for $Q_{ext,i}(D_{eff},\lambda)$ and $V$ into Eq. 8, we obtain

$$Vis = \frac{C A_{hp} v_T \delta t}{\sum_{i=1}^{N} 2A(i)}. \tag{10}$$

## 2.6 Measurement of snowflake and droplet terminal fall speed

The terminal fall speed of a snowflake is calculated using formula derived by Böhm (1998), namely

$$v_T = \frac{Re \cdot \eta}{2\rho_a} (\frac{\pi}{A})^{1/2}, \tag{11}$$

where $\rho_a$ and $\eta$ are the density and dynamic viscosity of air, respectively. The Reynolds number is define as a

$$Re = 8.5[(1 + 0.1519 X^{1/2})^{1/2} - 1]^2, \tag{12}$$

where $X$ is determined from atmospheric environment data and snow particle properties as

$$X = \frac{8mg\rho_a}{\pi\eta^2} (\frac{A}{A_e})^{1/4}, \tag{13}$$

where, $m$ (kg) is snow particle mass, $g$ is gravitational acceleration, $A_e$ (m$^2$) is the effective area normal to the flow and $A$ (m$^2$) is the circumscribed area around the snowflake that is estimated with a circle or ellipse using the major axis as a diameter.

Extensive studies have been performed to estimate the terminal fall speed of a raindrop as a function of diameter (Gunn and Kinzer, 1949; Rogers and Yau, 1989)

$$v_P = k_1 \left( \frac{D_{rain}}{20} \right)^{10k_2}$$

$$k_1 = 1.18 \times 10^6, k_2 = 2 \qquad \text{for } D_{rain} \leq 0.08$$

$$k_1 = 8 \times 10^3, k_2 = 1 \qquad \text{for } 0.08 \leq D_{rain} \leq 1.2$$

$$k_1 = 2.01 \times 10^3, k_2 = 0.5 \qquad \text{for } D_{rain} \geq 1.2 \tag{14}$$

where $D_{rain}$ is the diameter of a raindrop in cm and $v_P$ is the terminal fall speed in mm s$^{-1}$.

## 3  Methods

Two laboratory experiments were designed to calibrate the DEID and two field experiments were performed. The first lab experiment was used to calibrate the DEID and quantify its uncertainty in measuring hydrometeor mass. The second lab experiment was run in a wind tunnel to investigate the impact of environmental factors on the DEID performance. The first field experiment was conducted at the mouth of Red Butte Canyon at a location on the University of Utah campus that facilitated device debugging and enabled measurements to be more easily conducted throughout the winter. The second field study was a brief experiment conducted at Alta Ski Area's long-term monitoring site to provide an opportunity to validate the DEID against a weighing gauge (an industry standard method). Section 3.1 describes the DEID and its basic experimental setup that was used for each of the four experiments.

### 3.1  Overview of the DEID setup and image processing

The DEID consists of a hotplate with a feedback controller, a low-emissivity roughened aluminum top plate that is affixed to the top of the heater with thermal paste, and a thermal camera. The thermal camera used for all experiments is an uncooled microbolometer Infratec Vario HD 700 thermal camera with $1280 \times 960$ pixel resolution and sampling rates ranging from 2 to 30 Hz. The hotplate is a Systems and Technology International, Inc. HP-606-P that was used for all experiments. It is a custom unit with a heated area of 0.1524 m $\times$ 0.1524 m and a thickness of 0.0508 m. The hotplate is powered by a 120 V, 5-Amp supply and has a digital PID feedback control mechanism to control the plate temperature. The aluminum top plate is a 6061 alloy with a thermal conductivity, $k_{Al} = 205$ W m$^{-1}$K$^{-1}$, which was roughened using 2000 grit sandpaper in a linear motion across the plate yielding long straight grooves. A piece of Kapton® tape with high total hemispherical emissivity $\epsilon \approx 0.95$ is affixed to the top plate to measure the surface temperature using the thermal camera. Note that the thermal camera measures on the radiant surface or brightness temperature, which is only equal to the physical temperature of the substance for surfaces with $\epsilon = 1$.

For each experiment, the focus of the thermal camera was set manually using a high- and low-$\epsilon$ calibration sheet. The temporal and spatial variation of temperature across the hotplate is $\pm$ 0.1°C and $\pm$ 1°C respectively, which was measured using the thermal camera. The Infratec thermal camera writes out infrared binary (IRB) files that store the absolute temperature of each pixel. IRB files are converted into a gray-scale images, hence, the maximum temperature of the entire experiment has a 255 intensity value and the minimum temperature intensity is 0. The temperature to intensity conversion is linear. Analysis of the thermal images was performed using MATLAB®'s image processing toolbox where the linear interface between the hydrometeor and its background using a Sobel edge detection algorithm that computes the gradient of image intensity at each pixel within an image (Vincent and Folorunso, 2009). After applying the algorithm to each image, each pixel is assigned a value of either 1 for a hydrometeor or 0 for the background. In this work, we adopt 55/255 = 0.21 as the binary threshold. These processes were incorporated into a MATLAB script for tracking hydrometeor evaporation from the hotplate.

### 3.2 DEID laboratory-calibration experiments

To validate mass measurement, the DEID was placed in a 0.25 m per side open-topped cubic enclosure with an approximately zero wind speed of 0.02 m s$^{-1}$, a constant temperature of 20°C, and a constant relative humidity of 42%. Deionized water droplets of 0.02 g or 20 μL were applied to the hotplate 10 times using a pipette and allowed to evaporate. The plate used in this experiment had a thickness of $d_{Al} = 1$ mm and was maintained at a nominal temperature of 100°C. Two *k-type* thermocouples were affixed to the top and bottom of the aluminum plate using thermal paste to determine $T_b(t)$ and $T_p(t)$.

In order to validate droplet mass measurements both a micropipetter and gravity scale were used. The micropipette has an accuracy of 1.00/1.20 (%/μL). The gravity scale is a SARTORIUS model ENTRIS64-1S with a readability of 0.1 mg and repeatability (standard deviation) of 0.1 mg.

### 3.3 Environmental impacts on DEID mass measurement: wind-tunnel experiments

The DEID was placed in a custom built Engineering Laboratory Design Inc. wind tunnel. The tunnel consists of a settling chamber followed by a 6:1 2-D contraction that exits into the test section. The test section measures 2.7 m and has a 0.9 m × 1.2 m cross section. The upper surface of the test section articulates to allow adjustment of the axial pressure gradient. The maximum velocity in the test section is $\approx$ 12m s$^{-1}$ and the free-stream turbulence intensity is less than 0.4%. The following equipment was also used: a single straight-wire hot-wire anemometer system, an automated weather station, and a precision intravenous (IV) drip system for applying water droplets of fixed volume onto the hotplate (Figure 4). To ensure the IV produced a constant water-droplet volume discharge, the pressure head of the water bottle was maintained constant throughout the experiments. The metal plate was placed near the center of the wind tunnel test section and the thermal camera was deployed at a corner to minimize wind disturbance. Prior to the experiments, the hot-wire probe was calibrated in the tunnel.

The experiments were conducted with known 40 μL deionized droplet masses of water and ice for eight different wind speeds ranging from 0 to 10.3 m s$^{-1}$, five different hotplate surface temperatures between 80 and 110°C and five different relative-humidities between 36 and 92 %, in each case keeping the other two variables fixed. The humidity levels inside the wind tunnel were controlled using two humidifiers and a small fan to maintain a homogeneous distribution of humidity. To

monitor the uniformity of the spatial distribution of humidity, four humidity sensors at different vertical locations, 4, 11, 16, and 20 cm from the base of the wind tunnel, were placed around the metal plate. Each experiment was performed after reaching an approximately steady-state conditions for temperature, wind velocity and relative humidity.

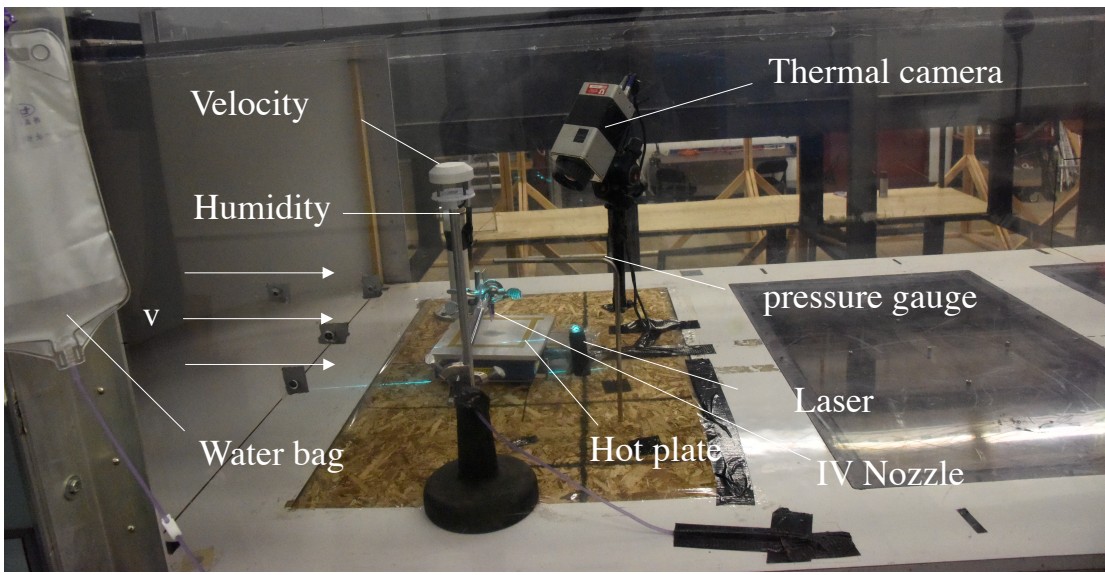

**Figure 4.** Photograph of the laboratory experimental set-up inside the wind tunnel used to validate the mass measurement of water droplets for various environmental conditions.

### 3.4 Field-validation experiments

Field experiments were conducted from 25 November 2019 through 16 April 2020 on the University of Utah campus at Red Butte Canyon (40.7686, -111.8263) and at the Alta Ski Area Collins Snow Study Plot (40.5763,-111.6383). At Red Butte Canyon, the DEID was mounted 1 m above the surface and at Alta Collins the DEID was mounted 1.25 m above the settled snow surface. At the Alta-Collins site, the DEID was collocated alongside instrumentation deployed at the long-running Collins Snow Study Plot (CLN), which is a well-protected snow-study site located at the upper terminus of Little Cottonwood Canyon, averaging 1300 cm of snowfall annually and 17.4 days with at least 25 cm of snow. The full record from CLN spans 41 years (January 1980–April 2021), and the last 21 seasons include a complete record of automated hourly precipitation observations (Alcott and Steenburgh, 2010). This site was chosen in part to avoid the additional measurement of windblown snow that would typically be lifted from exposed terrain features. However, we did not do anything to specifically avoid measuring lifted snow other than using this well-sheltered area along with keeping the plate surface elevated 1.25 m above the ground surface. Lifting the plate to this height significantly reduces wind-blown effects even in non-sheltered areas (e.g., Naaim-Bouvet et al. (2014)). Blowing snow is likely to have a distinct signature by way of particle clustering and size. In the current state, no distinction has

been made between the characteristics of free falling and lifted snow. If there is a flux of precipitation falling downward onto the plate, it will be measured whatever its origin.

The recommended operating range of the hotplate for field experiments is from 104°C to 106°C for ambient temperatures ranging from -20 to 20°C. At the Red Butte Canyon site, the hotplate was operated at 104°C for the entire experimental period between December 2019 and April 2020, with ambient temperatures ranging from -12 to 10°C. At the Alta-Collins site, the hotplate was operated at 106°C for the entire experimental duration between October 2020 and April 2021, with ambient temperatures that varied from -20 to 20°C. The average energy flux from the DEID at 0°C ambient temperature and with

wind speeds of 4.7 m s$^{-1}$ is 5581 W m$^{-2}$. No wind shield was placed around the DEID, such as those commonly used in precipitation gauge systems. The DEID was set to sample at 12 Hz at both study plots. An ETI Instrument Systems Noah-II precipitation weighing gauge was deployed 4 m from the DEID at the Alta Collins site and a wind shield was deployed around the ETI bucket to increase catchment efficiency. The ETI reported SWE measurements once every 1-hour. The resolution, threshold and accuracy of the ETI bucket is 0.254 mm, 0.254 mm, and ± 0.254 mm respectively.

Thermal imagery during the field experiments was acquired at a rate between 2 and 30 Hz depending on the the precipitation rate, although the data described here were primarily recorded at 12 Hz. At the Alta Collins site, 45718 snowflake images were considered for a 3-hour period for the analysis of histograms of mass, density, maximum diameter, equivalent diameter, complexity and aspect ratio.The instantaneous SWE rate and SWE accumulation were estimated every recorded frame, which was based on the total mass of snowflakes that had fallen on the hotplate in each frame. At the Red Butte Canyon site,

measurements were only during periods of continuous rain or snowfall. To produce snow and rain-droplet size distributions for each trial, 2000 snowflakes/raindrops were collected during continuous precipitation and the sample collection time varied from about 5 to15 minutes.

## 4   Results

### 4.1   DEID laboratory-calibration experiments

Ten 0.02 g, 20-µL water droplets were applied to the DEID heated plate using a pipette and the mass of each was determined using Eq. 4. The average of the DEID-computed water-droplet masses was 0.020 ± 0.0019 g.

Since these experiments were conducted in an enclosure where wind speeds were negligible, the effect of convective cooling on the mass calculation did not play a role.However, in the real natural environment (outside of the enclosure) winds can affect $T_p(t)$ (top side of the heated plate) but not $T_b(t)$ (bottom side of the heated plate that is always enclosed), in which case some

estimate must be made of convective heat losses from the plate due to external winds (Rasmussen et al., 2011). This issue can be addressed by replacing $T_b(t) - T_p(t)$ with $T_p(t) - T_w(t)$ and replacing $(k_{AL}/d_{AL})$ with an $(k_w/d_w)$ in Eq. 4 since the effect of convection losses due to ambient winds affects both $T_p(t)$ and $T_w(t)$ equally as shown in Fig. 5b.

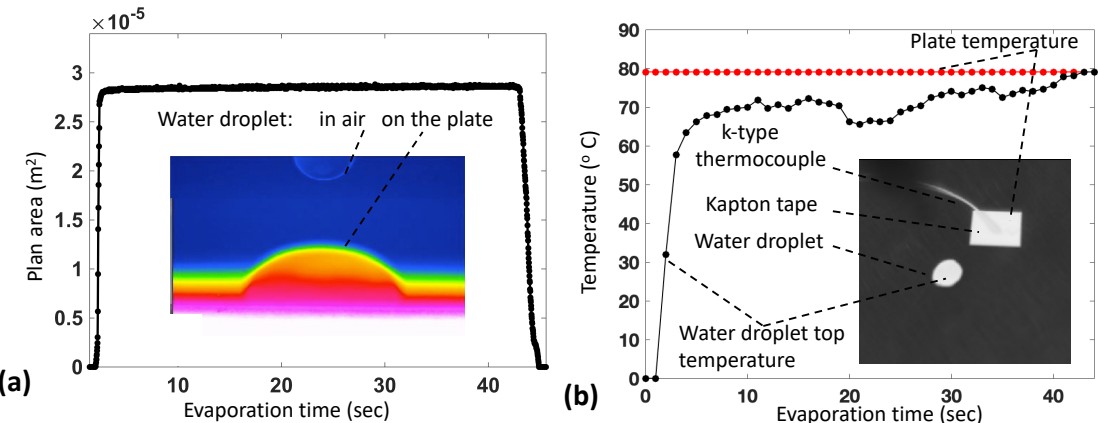

**Figure 5.** (a) Time series of the plan area of a water droplet during evaporation and a side-view temperature contour plot of a water droplet before and after impact on the DEID plate. (b) Time series of the temperature of plate and the top of the water droplet. The water droplet was applied with a pipettor seen as the bright round region. A rectangular piece of Kapton tape ($\epsilon \approx 0.95$) and a *k-type* thermocouple were used to calibrate the temperature of the plate and water droplet.

The justification for this approach is shown in Fig. 2. For quasi-steady conditions, conduction from the heated plate to a water droplet may be written as

$$\frac{T_b(t) - T_p(t)}{R_1} = \frac{T_p(t) - T_w(t)}{R_2}, \tag{15}$$

where $R_1 = (d_{AL}/k_{AL}A)$ is the thermal resistance across the aluminum plate and $R_2 = (d_w/k_w A)$ is the thermal resistance across water droplet, which must be determined through a calibration procedure in which known droplet masses are applied to the surface of the plate. Substituting the thermal resistance from Eq. 15 into Eq. 4 yields

$$c\Delta T \int dm + L \int dm = \int_0^{\Delta t} (k/d)_{eff} A(t)(T_p(t) - T_w(t))dt. \tag{16}$$

Note, in Eq. 16, we have replaced $k_w/d_w$ with a calibrated value $(k/d)_{eff}$. To determine $(k/d)_{eff}$, 0.02-g (20-μL) water droplets were individually applied ten times to the hotplate using a pipette. Eq. 16 was then rearranged to solve for $(k/d)_{eff}$. The results were averaged over the 10 samples yielding $(k/d)_{eff} = 7.006 \times 10^3$ W m$^{-2}$ K$^{-1}$.

With the derived value of $(k/d)_{eff}$, particle mass can be inferred from Eq. 16. This DEID-measured mass was compared against two high-accuracy standard methods: micropipetted droplets and weighed droplets using a gravimetric scale. Water-droplet volumes of 5, 10, 15, 20, 25, 30, 70, 80, 90, 100, 110, 120 μL were applied to the hotplate using a micropipette and weighed using a gravimetric digital scale. To ensure the complete discharge of the water droplet from the pipette during application to the hotplate and gravity scale, the pipette was placed very close to the plate/scale to maintain the continuity

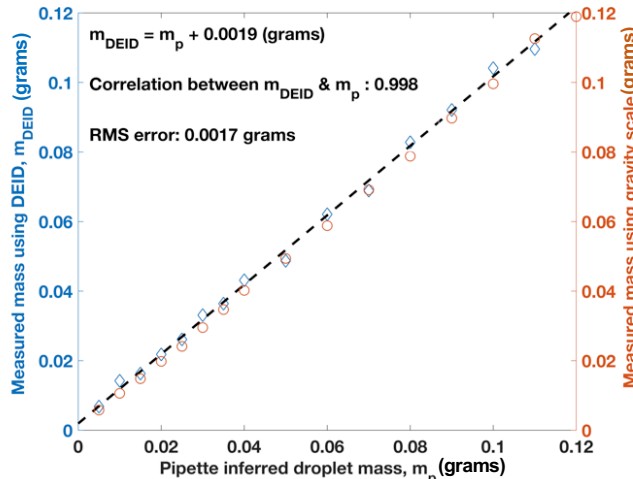

**Figure 6.** Correlation between water droplet mass measured from a pipette and that obtained using the DEID with the corresponding linear fit (coefficient of determination is 0.99). Mass of water droplet also measured using a weighing scale after apply water droplet on scale through pipette in similar way.

of discharge and the procedure was consistent for all trials. The mass measured by the DEID, pipette and gravity scale were averaged over three trials for each droplet water volume. Figure 6 shows that the correlation between DEID-measured droplet mass and pipette-inferred droplet mass is 0.99 with a root mean square error of 0.002 g. Furthermore, the correlation coefficient between the gravity-scale droplet mass and the pipette-inferred droplet mass is 0.99 with a root mean square error of 0.0018 g.

To validate the mass accumulation of multiple water droplets, experiments simulating rain were conducted by applying multiple droplets to the hotplate. Fifteen water droplets, each 0.04 g for a total 6 g measured with the gravity scale were applied to the hotplate one by one and measured with the DEID. The accumulated error was 0.023 g.

The DEID methodology for measuring the mass of ice particles was also evaluated. The primary difference between water and ice at 0°C is the added energy per kg required to overcome the latent heat of fusion $L_f$ prior to evaporation. To test this contribution, 0.04-g water droplets and 0.04-g ice particles made in a refrigerator in the laboratory, were applied to the hotplate and the average energy loss for an ensemble of 10 samples was computed using the right-hand-side of Eq. 17. The average energy required to evaporate the droplets was $101.4 \pm 3.2$ J and the average energy required to melt and evaporate the particles was $113.24 \pm 4.1$ J implying a mean latent heat of fusion of $2.96 \times 10^5$ J kg$^{-1}$, similar to the accepted value of $3.34 \times 10^5$ J kg$^{-1}$. Accordingly, to calculate the mass of the solid hydrometeors $L_v$ is replaced by $L_{eqv}$, we solve the following form of the energy balance equation for mass:

$$c\Delta T \int dm + L_{eqv} \int dm = \int_0^{\Delta t} (k/d)_{eff} A(t)(T_p(t) - T_w(t))dt \qquad (17)$$

where $L_{eqv} = L_v + L_f$.

### 4.2 Environmental impacts on DEID mass measurement: wind-tunnel experiments

To determine how wind speed affects DEID mass measurement, all environmental parameters except velocity were maintained approximately constant in the wind tunnel while the wind speed was varied from 0.5 m s$^{-1}$ to 10.3 m s$^{-1}$. Water droplet-experiments were performed in the wind tunnel with wind speeds of 0, 0.6, 1.5, 3.5, 5.5, 7.2, 8.84, and 10.3 m s$^{-1}$. For each trial, 40-$\mu$L (0.04 g) water droplets were placed on the heated plate and three trials were performed for each wind speed. Results are summarized in Table 1. The measured total energy loss from the plate for each trial was approximately constant and independent of ambient wind speed. averaging $100.77 \pm 4.72$ J for an average measured DEID mass of $0.044 \pm 0.0019$ g.

To investigate the effects of humidity variability, the wind tunnel was set at 37%, 50%, 70%, 80%, and 92% relative humidity with a measured wind speed of approximately zero (0.02 ms$^{-1}$). The temperature of the plate was set to 100°C and again 40-$\mu$L, or 0.04 g water droplets were applied to the heated plate. Three trials were performed for each level of humidity and measurements were taken of the total energy loss for each trial. The results summarized in Table 2 indicate a negligible dependence. The average measured energy loss for all humidity levels was $99.96 \pm 4.42$ J for an average measured mass of $0.044 \pm 0.0016$ g.

The final test was to vary the DEID plate temperature between 80°C and 110°C, while maintaining a fixed wind speed of 0.02 m s$^{-1}$ and relative humidity of 37%. The water-droplet experiments were performed with surface plate temperatures of: 80°C, 90°C, 100°C, 105°C and 110°C. Results summarized in Table 3 show a measured average energy loss of $101.78 \pm 4.8$ J and a measured average mass of $0.044 \pm 0.0018$ g.

The conclusion is that DEID measurements are highly insensitive to environmental conditions and device settings unlike prior hotplate devices that require detailed ambient measurements and corrections to obtain precise measurements of precipitation rate (Rasmussen et al., 2011; Thériault et al., 2021) .

**Table 1.** Mean and standard deviation of mass ($m$) and energy loss ($E$) per droplet from the hotplate measured using the DEID methodology for eight different wind speeds ($WS$) for 0.04 g water droplets.

| $WS$ (m s$^{-1}$) | E (J) | m (g) |
|---|---|---|
| 0.02 | $101.4 \pm 3.71$ | $0.044 \pm 0.0015$ |
| 0.6 | $98.8 \pm 2.81$ | $0.043 \pm 0.0011$ |
| 1.5 | $95.8 \pm 6.21$ | $0.042 \pm 0.0025$ |
| 3.5 | $100.3 \pm 5.18$ | $0.044 \pm 0.0021$ |
| 5.5 | $107.9 \pm 4.61$ | $0.047 \pm 0.0019$ |
| 7.2 | $105.5 \pm 3.96$ | $0.046 \pm 0.0016$ |
| 8.84 | $102.1 \pm 6.09$ | $0.045 \pm 0.002$ |
| 10.3 | $95 \pm 5.21$ | $0.042 \pm 0.002$ |
| | $100.77 \pm 4.72$ | $0.044 \pm 0.0019$ |

**Table 2.** Average and standard deviation mass ($m$) and energy loss ($E$) per droplet from the hotplate measured using the DEID for numerous relative humidity values ($RH$) for 0.04 g water droplets.

| $RH(\%)$ | $E$ (J) | $m$ (g) |
|---|---|---|
| 37 | $102.2 \pm 3.86$ | $0.045 \pm 0.0012$ |
| 50 | $100.6 \pm 5.86$ | $0.044 \pm 0.0021$ |
| 70 | $98.6 \pm 4.21$ | $0.043 \pm 0.0015$ |
| 80 | $103.8 \pm 5.21$ | $0.046 \pm 0.0018$ |
| 92 | $94.6 \pm 2.96$ | $0.042 \pm 0.001$ |
| | $99.96 \pm 4.42$ | $0.044 \pm 0.0016$ |

**Table 3.** Average and standard deviation mass ($m$) and energy loss ($E$) per droplet from the hotplate measured using the DEID for different hotplate temperatures for 0.04 g water droplets.

| $T(°C)$ | $E$ (J) | $m$ (g) |
|---|---|---|
| 80 | $99.6 \pm 6.23$ | $0.044 \pm 0.0023$ |
| 90 | $102.8 \pm 5.2$ | $0.045 \pm 0.0019$ |
| 100 | $98.8 \pm 4.28$ | $0.043 \pm 0.0015$ |
| 105 | $103.6 \pm 3.21$ | $0.045 \pm 0.0011$ |
| 110 | $104.1 \pm 5.1$ | $0.046 \pm 0.0018$ |
| | $101.78 \pm 4.8$ | $0.044 \pm 0.0018$ |

## 5  Field validation

Field experiments conducted at Red Butte Canyon and Alta-Collins Snow Study Plot provided DEID measurements of SWE accumulation, snow accumulation and snow density measurements, and particle attributes that could be compared with independent sensors. An example of the DEID thermal imagery data acquired at Alta Collins is presented in Fig. 7 which shows how binary thermal imagery of snowflakes, can be converted into an effective circular diameter $D_{eff}$ and a maximum effective diameter $D_{max}$.

Fig. 8 shows probability distributions for snowflake mass, density ($\rho_s$), effective circular diameter ($D_{eff}$), maximum effective diameter ($D_{max}$), complexity, and the ratio of melted diameter ($D_{mel}$) to effective circular diameter ($D_{eff}$).

For systematic and random error analysis, 45718 snowflakes have been considered, which were collected during an $\approx$ 6-h period during field experiments at Alta Collins on 15 April 2020. During this period, a wide range of precipitation rates ranging from 0.001 to 16 mm hr$^{-1}$, were observed. Direct measurements made by the DEID consist of: area, temperature, and the evaporation time of snowflakes. The percent error in the area, temperature, and evaporation time for all observations is 1.0 %, 0.3 %, and 1.0% respectively. The percent error in the calibration constant $(k/d)_{eff}$ is 1.0%. The percent error in derived quantities (using a standard propagation of uncertain analysis) such as equivalent diameter, particle complexity, mass, density, visibility, SWE, and snow height are 0.5%, 2.0%, 3.3%, 4.8%, 3.3%, 5.3%, and 8.1%, respectively. The probability

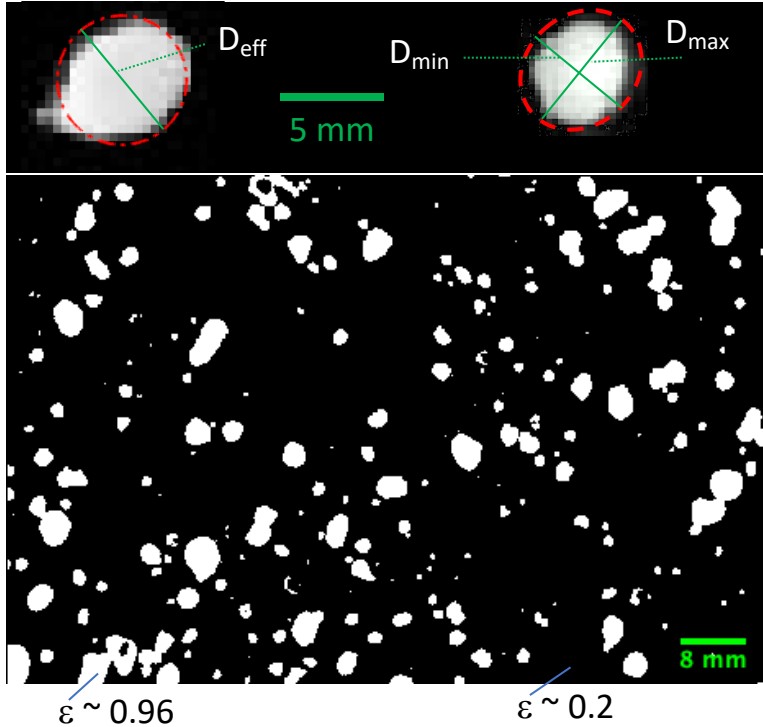

**Figure 7.** (Bottom panel) Black and white binary thermal images of snowflakes in various stages of melting and evaporation on the DEID heated plate observed at Alta. (Top panel) A close-up image illustrating the definitions of $D_{eff}$, $D_{min}$ and $D_{max}$. $\epsilon_w$ of snow and Aluminum are noted.

of subsequent hydrometeors falling on top of one another before complete evaporation of the initial hydrometeor depends mostly on the following parameters: precipitation rate, hotplate temperature, evaporation time, snowflake type, and density. To calculate the coincidence probability, the same data introduced above were considered with a given hotplate temperature of 104°C. When compared to a typical evaporation cycle for a single frozen hydrometeor, overlapping is indicated by a significant decrease in temperature and increase in area within a normal cycle of evaporation. By applying these conditions, the probability

of coincidence was calculated. A second method takes into account the size distribution, which provides a vertical structure of hydrometeors based on precipitation rate. An overlap is counted if the evaporation time of any hydrometeors is greater than the average time between two consecutive hydrometeor in the vertical direction. Using these two methods, negligible overlaps were observed for a precipitation rate of ≈1 mm hr$^{-1}$, and a maximum of 4.9 % coincidence probability was observed during the highest SWE rate 15.6 mm hr$^{-1}$. Note that even during instances of overlap, in contrast with optical disdrometers, the

DEID does not 'lose' measurements of the primary hydrometeor quantity amount, in this case mass. The DEID provides a combined mass as discussed in Appendix A1. While total mass estimation is unaffected, individual particle calculations such

as mass, size, and density are. For data where overlap is identified, these measurements are not considered in the probability and size distributions, etc. presented herein.

The following are median values with lower and upper quartiles for the above parameters: mass = 0.46 [0.20 1.18] mg; $D_{eff}$ = 1.73 [1.37 2.42] mm; $D_{max}$ = 1.89 [1.26 2.84] mm; $\rho_s$ = 92 [61 120] kg m$^{-3}$; complexity = 1.41 [1.25 1.68]; $D_{mel}/D_{eff}$ = 0.61 [0.49 0.73]. In general, representative parameters of snowflake mass, size, density, complexity, ratio of $D_{mel}$ to $D_{eff}$ acquired during the storm shown were highly variable; the mean mass was $1.80 \pm 9.04$ mg and mean density was $92 \pm 42$ kg m$^{-3}$. The most likely value of $D_{eff}$, $D_{max}$ and $\rho_s$ were 1.34 mm, 1.58 mm and 97 kg m$^{-3}$, respectively, which is consistent with past measurements at the same site (Garrett et al., 2012; Alcott and Steenburgh, 2010).The distribution of the ratio of $D_{mel}$ to $D_{eff}$ is slightly positively skewed with a skewness 0.10 and kurtosis 3.24). Also, the typical value of complexity was 1.22, indicating the predominance of rounded/rimed snowflakes. Using a plate temperature 104 °C, a thermal camera sample frequency of 12 Hz, and for 45718 snowflakes, the median hydrometeor evaporation time including lower and upper quartiles was 2.41 [1.25, 5] sec. Hence, this range of time scales minimizes uncertainty in the measurement of all types of hydrometeors at a given hotplate temperature and frame rate.

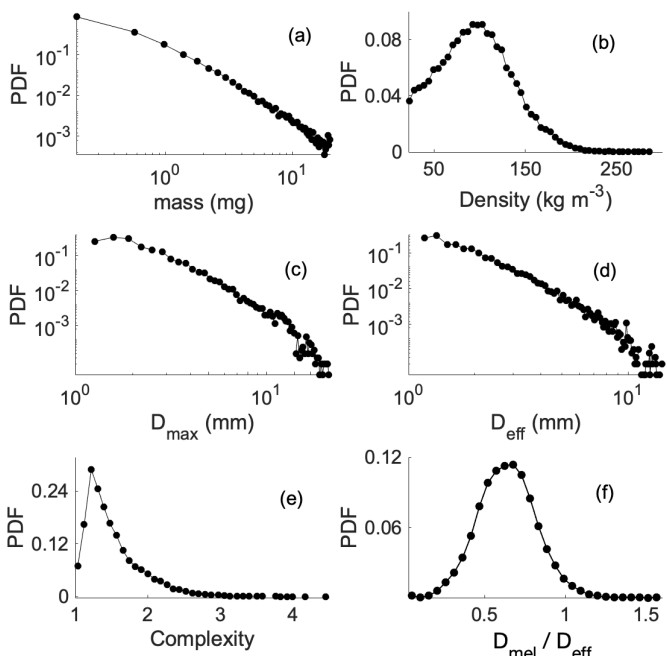

**Figure 8.** Distributions of snowflake characteristics measured using the DEID at the Alta-Collins Snow Study Plot from a sample of 45718 snowflakes. (a) Mass, (b) density, (c) maximum effective diameter $D_{max}$, (d) effective circular diameter $D_{eff}$, (e) complexity, and (f) ratio of melted spherical diameter and effective circular diameter.

As part of the validation exercises in this study, the DEID was deployed alongside a Multi-Angle Snowflake Camera (MASC) (Garrett et al., 2012) at the Red Butte Canyon site. The MASC is composed of three high-speed optical cameras that image individual snowflakes as they fall through the field of view. The MASC can be used to obtain accurate estimates of snowflake sizes. A one-hour period of measurements was used for comparing MASC and DEID measurements of hydrometeor maximum dimension $D_{max}$. The median values with lower and upper quartiles from the DEID and MASC are $D_{max} = 2.77$ [1.84 4.38]

360   mm and $D_{max} = 2.90$ [1.93 4.89] mm respectively. Hydrometeor maximum-dimension PDFs from both instruments are given in Fig. 9. The results indicate that the snowflake distributions measured by the two instruments are very similar. A more extensive comparison between the MASC and DEID is addressed in Rees et al. (2021).

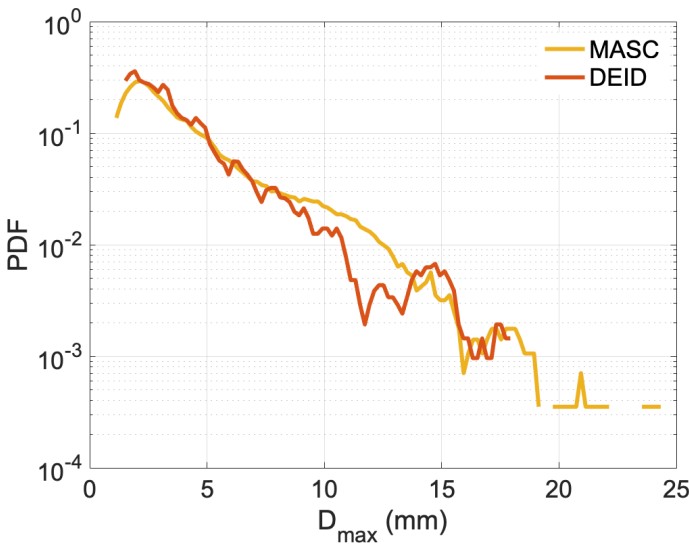

**Figure 9.** PDF of $D_{max}$. 2268 snowflakes were observed using the DEID and 2093 snowflakes were observed using the MASC during a one-hour period on 16 Jan 2020 at Red Butte Canyon.

### 5.1   Validation of SWE rate measurements

An approximately 1-hour long time series of raw (12 Hz) and 15-sec averaged SWE rate data taken at Alta Collins is shown

in Fig. 10. Here, periods with a SWE rate of less than 0.001 mm hr$^{-1}$ or characterized by small hydrometeors with $D_{eff} <$ 0.2 mm are assumed to correspond with no snowfall. Broad variability indicating fine-scale storm structure is observed. For example, during the 15 April 2020 snowstorm shown, SWE rate rapidly changed from 0.1 to 40 mm hr$^{-1}$ within 5-min. Such detail cannot be identified using traditional snow-accumulation measurement techniques.

    DEID SWE accumulation was compared with an industry standard ETI Noah-II precipitation weighing gauge. Both instru-

ments were deployed within 4 m of one another at the Alta-Collins site. The DEID sampling frequency was set at 12 Hz,

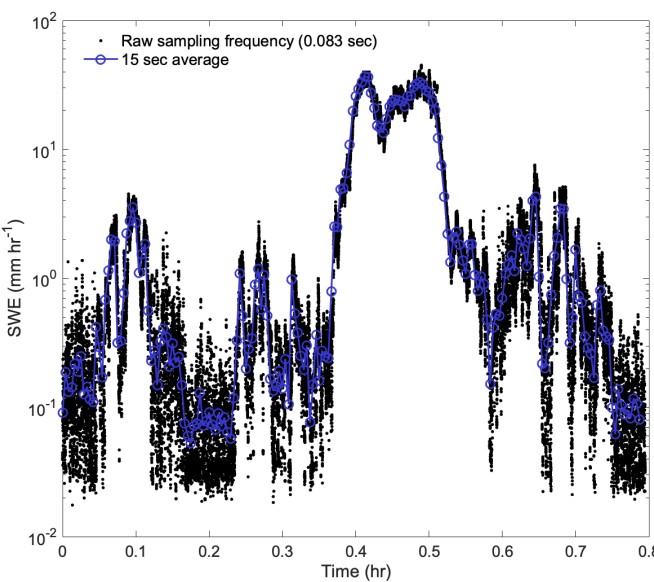

**Figure 10.** Time series of SWE rate measured at a sampling frequency of 12 Hz (black dots) and averaged over a 15 sec period (blude dots) measured on 15 April, 2020 at the Alta-Collins site.

while data from the ETI were reported every hour. Data were collected from 0000 UTC 15 April to 1600 UTC 15 April 2020. Accumulated SWE integrated over five minute intervals is plotted against the ETI data in Fig. 11. DEID SWE accumulation observations match those from the ETI gauge to within $\pm$ 6% over the 16-hour measurement period. The DEID SWE accumulation is slightly higher than the ETI because the minimum resolution of ETI is 0.254 mm whereas the minimum DEID resolution is 0.001 mm. To determine a thermal camera frame rate that would capture the widest possible range of hydrometeor types, an experiment was performed during a snow event at Red Butte Canyon on 25 March 2020. The thermal camera was operated at a frequency of 60 Hz with the plate temperature set to 104°C. The total mass of hydrometeors was estimated using two different algorithms one that estimates the total mass in each frame using the energy balance equations and second that computes the mass of each particle following them across a series of frames. The total mass of hydrometeors that fell on the hotplate within half an hour was calculated using sampling frequencies of 1, 2, 3, 6, 10, 12, 15, 20, 30, and 60 Hz. Using the frame-by-frame method the calculated total mass at 12 Hz frequency was found to be 99.8% of the total mass calculated at 60 Hz. Hence, the 12-Hz frame-by-frame method was used for SWE accumulation calculations. Using the particle-by-particle method, the calculated total mass at a 12-Hz frame rate was found to be 94.79% of the total mass calculated at 60 Hz. While sampling at 60 Hz could be done, it is less practical operationally. For a $\approx$1.2 Mpixel camera resolution, the processing time for each frame is approximately 0.015 sec. The average size of the dataset for a one-hour period is 1.3 Gb and the associated processing time is $\approx$11 minutes. Selecting a frame rate of 12 Hz, in part, assures that the DEID can operate as a real-time

instrument. Hence, the 12 Hz represents a cost benefit balance between accuracy of the measurement and time and storage costs.

Figure 11 also suggests the DEID can faithfully measure snow density throughout a 16-hour storm. Low-density snow (48 kg
m$^{-3}$) transitioned to higher-density snow (176 kg m$^{-3}$) before ending with slightly lower density (92 kg m$^{-3}$) accumulations. The ability of the DEID to capture this complex density layering is critical to applications such as avalanche forecasting.

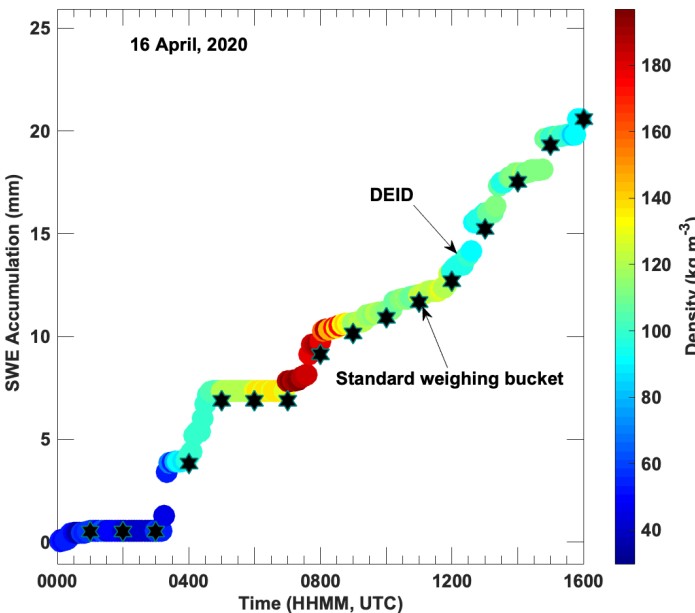

**Figure 11.** Time series of SWE accumulation measured using the DEID and ETI gauge along with DEID-measured snow density. The data were acquired at Alta Collins on 16 April 2020. Each DEID data point represents a 5-min average.

## 5.2    Snow characterization and density measurements

Figure 12 shows four different types of snowflakes inferred using the DEID at the Red Butte Canyon site and their estimated mean densities. Images of snowflakes on the hotplate are generally well-separated from each other, allowing for calculation of
individual mass, size, and density. Figure 12a and b show snow particles consisting of aggregates with mean densities of 95 ± 6 kg m$^{-3}$ and 82 ± 11 kg m$^{-3}$, respectively. Figure 12c shows dense graupel with a mean density of 260 ± 21 kg m$^{-3}$ and Fig. 12d shows snow particles with a wide range of sizes with a low mean density of 42 ± 26 kg m$^{-3}$. Time series of key bulk precipitation quantities measured at Alta Collins are shown in Fig. 13. The data include 1-min averaged visibility, density, SWE rate, and $PI_{snow}$. Averaged over the one hour shown, the estimated density was 124 ± 54 kg m$^{-3}$ and the lowest
visibility measured was 0.415 km, which was associated with a 5-minute period (21 to 26 minutes) of particularly heavy snow

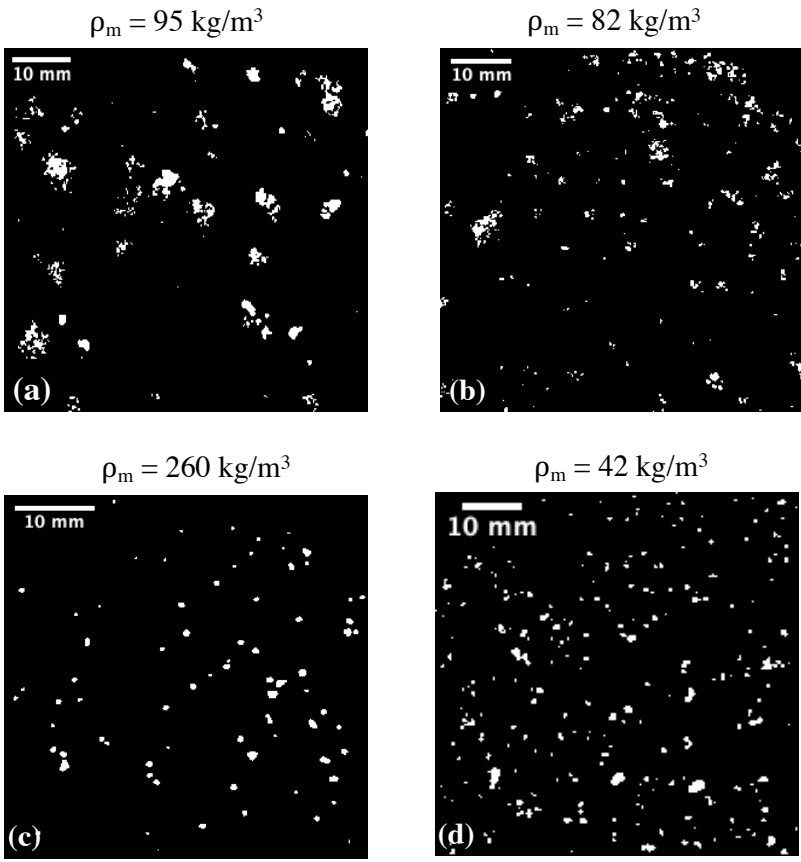

$\rho_m = 95$ kg/m$^3$          $\rho_m = 82$ kg/m$^3$

$\rho_m = 260$ kg/m$^3$          $\rho_m = 42$ kg/m$^3$

**Figure 12.** Image of snow particles measured by the DEID at Red Butte. The mean density calculated from the mass and effective spherical volume is (a) 95 kg m$^3$. (b) 82 kg m$^3$ (c) 260 kg m$^3$ and (d) 42 kg m$^3$.

fall. The heavy snow was followed by a period where the visibility increased to than 5.0 km when snowfall was light (from 41 to 45 minutes).

## 6   Scientific application: size distributions

One of the first studies to quantify rain-droplet size distributions was performed by Wiesner (1895) who measured individual
raindrop size after it had fallen onto a piece of plotting or filter paper. Here, we compare DEID measured size distributions with canonical results obtained previously by (Marshall and Palmer, 1948) for rain and (Gunn and Marshall, 1958) for snow that are used extensively in the atmospheric sciences literature. A key feature of these results is an exponential tail that is less steep with increasing precipitation rate and a constant intercept independent of rate at a diameter near zero for rain and greater than

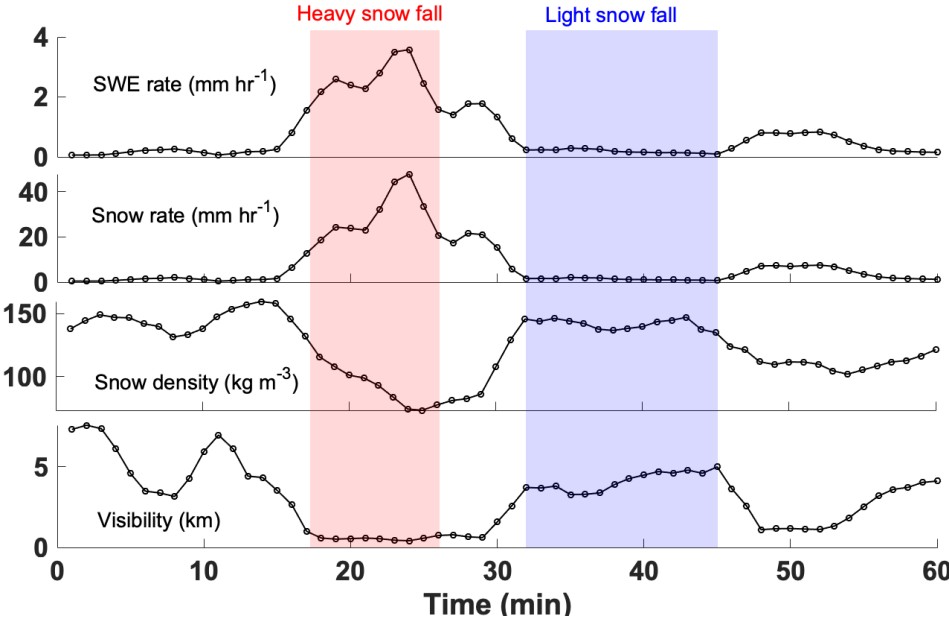

**Figure 13.** Time series of 1 min average of snow water equivalent (SWE), snow precipitation rate ($PI_{snow}$, mean snow density and visibility obtained at Alta-Collins on 15 April 2020).

zero for snow. It has been shown that the functional form of the distribution can be arrived at by considering growth through hydrometeor collisions that 'transport' particles in and out of successively larger size bins as balanced by an increasing terminal fall speed with size (Garrett, 2019).

Accurate ground based measurement of precipitation size distributions either relies on particle-by-particle measurement using optical devices or is inferred from bulk measurements using for example a radar. In either case, accuracy of both the direct measurements and any assumptions can be adversely affected by high winds and turbulence and, for snow, an unknown density (Thériault et al., 2012). The DEID, however, being simply a horizontal flat plate, is not expected to suffer from collection inefficiencies, except for minimal interference with falling hydrometeors by the thermal camera.

## 6.1 Rain

A consideration for measurement of size however is that the area of raindrops is rapidly distorted upon impact (Parsakhoo et al., 2012). With the DEID, greater deformation in size was observed for larger droplets. Therefore, we use an effective spherical diameter inferred from the mass measurement and density of water. We focus on three rain events occurring on three different days during the field experiments conducted at Red Butte Canyon. For each day, a sample of $\approx 2000$ rain droplets is taken for size distribution analysis. To obtain concentrations (number of rain drops per unit air volume) an effective volume of air was estimated from the product of the sampling area of the hotplate and an effective vertical distance in sample collection time.

The effective vertical distance is estimated using the product of the mean fall speed and the sampling duration. The terminal
fall velocity of the raindrops was calculated using Eq. 14 and an average velocity taken over 2000 raindrop samples is used
to calculate an effective vertical height. The size distribution of raindrops for three different precipitation rates is given in Fig.
14; the average $N_0$ (y-axis intercept at $D_{rain} = 0$) is $8.13 \times 10^3$ m$^{-3}$ mm$^{-1}$, which is well matched to those value obtained by
Marshall and Palmer (1948) for all rain rates.

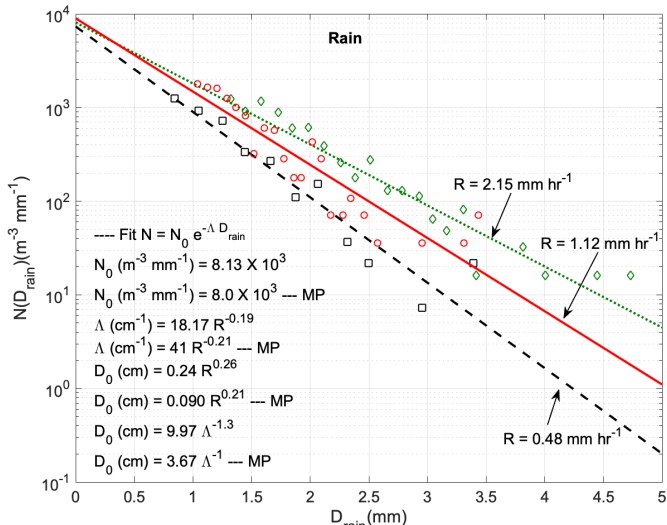

**Figure 14.** Size distributions of raindrops for different fall rates measured in Red Butte Canyon. Approximately 2000 raindrops were considered in each case binned in increments of $\approx 0.2$ mm. Distribution of raindrops (points) on a plot of log number vs diameter of raindrops
$(D_{rain})$, fitted using N($D_{rain}$) = N$_0$ e$^{-\Lambda D_{rain}}$ (lines). R is SWE rate of rainfall ($S\dot{W}E$) and D$_0$ is median of diameter of raindrops. Fitted
results are compared with Marshall and Palmer (1948), abbreviated as MP.

## 6.2 Snow

As described above, snowflake sizes ($D_{eff}$) can be directly obtained from area measurements made by the DEID. Size distributions for ensembles of $\approx 2000$ snowflakes binned in 0.2 mm increments are presented in Fig. 15 as $\log N(D_{eff})$ versus
$D_{eff}$. The plots show that the data are well described by exponential fits of the form $N(D_{eff}) = N_0 \, e^{-\Lambda D_{eff}}$ for $D_{eff} > 1$
mm. The mean snow density and precipitation rate was averaged over 2000 snowflakes.

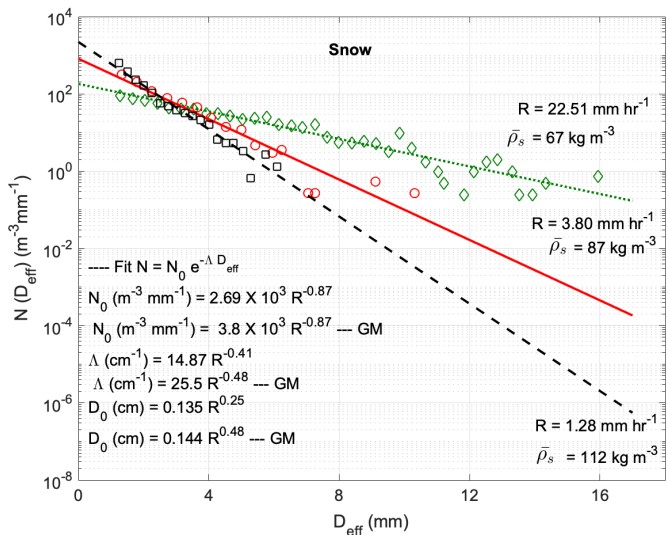

**Figure 15.** Size distributions of snow particles for different SWE rates ($S\dot{W}E$). Approximately 2000 snowflakes were considered in each case with bin sizes of 0.2 mm. The mean snow density is taken by averaging over $\approx$ 2000 snowflakes. Distribution of snowflakes (points) on a plot of log number vs effective circular diameter ($D_{eff}$), fitted using $N(D_{eff}) = N_0\, e^{-\Lambda D_{eff}}$ (lines). R is SWE rate of snowfall ($S\dot{W}E$) and $D_0$ is median of $D_{eff}$. Fitted results are compared with Gunn and Marshall (1958) abbreviated as GM.

## 7    Conclusions

We have described a novel ground-based thermal and optical instrument, the Differential Emissivity Imaging Disdrometer (DEID), the first device shown to be capable of accurately measuring the physical properties of individual hydrometeors, including paticle mass, density, and size. This is the first particle-by-particle device capable of measuring mass, density and size of hydrometeors, and of integrated measurements widely used in the meteorological and atmospheric sciences community, that has been shown to perform with high accuracy.

The DEID concept is simple. It consists of a heated metal plate with a low infrared emissivity top surface viewed by a thermal camera. The heat loss from the plate required to melt and evaporate high emissivity solid and liquid hydrometeors is estimated using a thermal camera. Finally, the heat loss is converted into a mass using via a control volume-based energy budget computed for each hydrometeor. The camera's sampling frequency and the resolution of the images determine the measurement error. In this work we used a thermal camera with a resolution of $1280 \times 960$ pixel for which the minimum size

accepted by the DEID is 1 pixel, which is 0.2 mm. Furthermore, the DEID can measure precipitation rates with a sampling frequency of 12 Hz ranging from 0.001 to 200 mm hr$^{-1}$. The accuracy of the measurements is partially an inverse function of plate area due to errors associated with sampling statistics Rees and Garrett (2021).

In laboratory measurements, the DEID was found to be highly insensitive to environmental conditions including wind speed, temperature, and humidity, Notably, in contrast with previous precipitation-gauge instruments based on a hotplate concept

(Rasmussen et al., 2011), the DEID measurement principle does not depend on wind speed as the mass calculation depends on the temperature difference between the hydrometeor and hotplate surface.

The DEID performed well in preliminary field experiments conducted at two different sites. Measurements taken during a snowstorm demonstrated the instruments' ability to observe precipitation rates and snow densities at unprecedented sampling frequencies while maintain fidelity to within 6% of the industry standard ETI weighing device. Size distributions obtained

during a rain and snow events are consistent with those published previously in the literature. While these early results need to be validated under a wider range of conditions, they show high potential to provide important new precipitation data streams to meteorologists, hydrologists, and avalanche forecasters.

## Appendix A: Heat loss calculations

### A1 Calculation of convective heat loss

For calculation of convective and radiative heat loss during evaporation of a water droplet, 40 μL of water was applied to the hotplate using a micropipette. The total energy required to evaporate 40 μL or 0.04 g of water at $100°C$ can be estimated using the following equation

$$Q_{total} = mL_v + mc\Delta T. \tag{A1}$$

$Q_{total}$ is total energy required to evaporate the droplet, which is 103.8 J using $L_v = 2.26 \times 10^6$ J kg$^{-1}$, $c = 4.182 \times 10^3$ J kg$^{-1}$

K$^{-1}$ and $\Delta T = 80$ K. The convective heat loss during evaporation of a water droplet is

$$Q_c = \int_0^{\Delta t} h_c A \Delta T dt, \tag{A2}$$

where $Q_c$ is the convective heat loss and $h_c$ is the convective heat-transfer coefficient. The heat-transfer coefficient is calculated using (Kosky et al., 2013)

$$h_c = \frac{K}{D} 0.0158 (Re)^{0.8}, \tag{A3}$$

where $K$ is thermal conductivity of air, $D$ is diameter of water droplet, which is approximately constant during evaporation, and $Re$ is the Reynolds number that is calculated using following equation

$$Re = \frac{VD}{\nu}. \tag{A4}$$

Here, $V$ is the air velocity and $\nu$ is kinematic viscosity of air. The calculated convective heat loss for a given area ($5.83 \times 10^{-5}$ m$^2$), velocity (3.5 m s$^{-1}$) and diameter of the water droplet (0.0086 m) is 1.04 J.

## A2 Calculation of radiative heat loss

The radiative heat loss is estimated using the following equation

$$Q_R = \epsilon_w \sigma b \int\limits_0^{\Delta t} A(t)(T_w^4(t) - T_{air}^4)dt, \tag{A5}$$

where, $\epsilon_w$ (0.98) is emissivity of water, $b$ (0.66) is the view factor between water surface air surrounding and $T_{air}$ is ambient air temperature that is 25°C. Calculated radiative heat loss using A5 is 1.09 J.

## Appendix B: Cleaning of the hotplate

Dust storms can leave static residue on the hotplate after evaporation that is imaged by the thermal camera. This residue produces a bright visual signature on the hotplate surface that is seen by the thermal camera. To regain an accurate measurement, the dust residue needs to be removed (cleaned) from the hotplate surface. The following procedure is typically used to clean the hotplate: (1) Manually cleaning by placing fresh snow onto the plate and then wiping with a dry clean cloth. (2) Self-cleaning during snow events – the hotplate is briefly turned off remotely at the beginning of the storm and then turned back on after an accumulation ($\approx$ 2 mm) of fresh snow on the plate. It is common for some ($\approx$ 0.001% area of the hotplate) bright spots (residue) to remain on the hotplate surface throughout the entirety of a storm. Typically, these bright spots can be removed computationally when using either the frame-by-frame or particle-by-particle methods discussed in the main text. In the frame-by-frame method, the total mass due to the residue was subtracted in each frame and the total area of residues was subtracted from the hotplate area. In the particle-by-particle method, all hydrometeors must complete the cycle of evaporation where the area of hydrometeor must be zero at the beginning and end of the evaporation. But residues do not evaporate and change area like hydrometeors; hence, residues were not counted and the hotplate area was reduced by subtracting the total area of residues.

## Appendix C: Bouncing of snow from the hotplate and catchment efficiency of the DEID

Snow particles bouncing from the hotplate are a function of two-time scales, which are the contact time between the plate and snow particle and the melting time of the bottom layer of snow particle. There is a competition between contact time and melting time and contact time decreases with the increasing density of the snow particle. However, melting time increases with the increasing density of snow particles. For a given density of snow particle (74 kg m$^{-3}$), contact time is $\mathcal{O}$ (10$^{-1}$ sec), and the melting time of a 100-$\mu$m thick layer of snow is $\mathcal{O}$ (10$^{-3}$ sec). When the snow particle melts, the normal reaction force from the surface to the snow particle is weekend. A roughened plate and surface tension between plate and water layer help to hold the snow particle after impact along the surface of the heated plate.

*Code availability.* The data processing codes are protected through a patent.

*Data availability.* The datasets are available upon request.

*Author contributions.* DKS contributed to experiment design and setup, data collection, analysis, and writing of the document. SD contributed to experiment design and setup as well as data collection and review of the document. ERP and TJG contributed to experiment

design and setup, project advising, as well as writing and editing of the manuscript. In addition, TJG acted as project lead.

*Competing interests.* The DEID is protected through a patent (WO2021108776) co-authored with DKS, ERP, and TJG and is commercially available through Particle Flux Analytics, Inc. TJG is a co-owner 260 of Particle Flux Analytics, Inc. which has a licence from the University of Utah to commercialize the DEID.

*Acknowledgements.* We thank Karlie Rees and Trent Meisenheimer of the University of Utah for their important contributions to the exper-

iments and analysis. The authors also wish to thank Dave Richards and the Alta Ski Patrol for their help and contributions that were critical to facilitating the field validation study at Alta Ski Area.

We thank Allan Reaburn and his colleagues at Particle Flux Analytics for their key contributions in the development of DEID. This product is protected and commercially available through Particle Flux Analytics, Inc. Co-author Tim J. Garrett is a co-owner of Particle Flux Analytics.

This research was accomplished via support from the U.S. National Science Foundation grant number PDM-1841870, the U.S. Department of Energy, Office of Science, grant number SC-0017168 and the Transportation Avalanche Research Pooled Fund Program (Colorado Department of Transportation) Study No. 5337-20-09.

The views expressed are those of the authors, and do not represent any accuracy, liability, warranty and do not necessarily state or reflect those of the United States Government or any agency thereof.

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
