# Peer review of "A differential emissivity imaging technique for measuring hydrometeor mass and type"

_Atmospheric Measurement Techniques, 2021_

## Author Comment (AC1)

**Reviewer 1 comments and response**

We appreciate the reviewers insightful comments and criticisms. We have tried to address them all carefully below in a point-by-point manner and believe the paper has been improved as a result.

**Review Comment 1:** *There are a number of shortcoming that need to be addressed before this paper can be published. The most significant being the lack of a comprehensive error analysis the documents the source of systematic and random errors and then propagates these into the derived quantities that are being highlighted, i.e., equivalent diameter, particle complexity, density, mass, visibility, SWE, etc. There are many potential sources of uncertainty that were mentioned but no quantitative estimates given. This is unacceptable for an instrumentation paper. One of the uncertainties that is given very short shrift concerns the probability that two more snowflakes will be imaged together, not because they are aggregating when they fall but because one fell one top of the other. A very brief comment is made that under one condition, out of a 1000 images, only 5 were touching. Figure 7 belies that statement since there are many fewer than 1000 particles and I count more than 10 that are touching. Given the long times needed to evaporate ice crystals (see my next enumerated issue), 30-60 seconds, under even modest precipitation rates the probability must be moderately high that as one crystal melts/evaporates, another will fall on top of it. This situation is not addressed but a very simple calculation needs to be made, similar to what is done with other optical spectrometers, to estimate the coincidence probability for different size distributions and precipitation rates.*

**Response:** We agree with the referee that a comprehensive error analysis is important to address in the paper. For systematic and random error analysis, 45718 snowflakes have been considered, which were collected during an approximate 6 h period during field experiments at Alta Collins on 15 April 2020. During this period, a wide range of precipitation rates ranging from 0.001 to 16 mm hr$^{-1}$, were observed. Direct measurements made by the DEID consist of: area, temperature, and the evaporation time of snowflakes. The percent error in the area, temperature, and evaporation time for all observations is 1.0%, 0.3%, and 1.0% respectively. The percent error in the calibration constant $(k/d)_{eff}$ is 1.0%. The percent error in derived quantities (using a standard propagation of uncertain analysis) such as equivalent diameter, particle complexity, mass, density, visibility, SWE, and snow height are 0.5%, 2.0%, 3.3%, 4.8%, 3.3%, 5.3%, and 8.1%, respectively.

The probability of subsequent hydrometeors falling on top of one another before complete evaporation of the initial hydrometeor depends mostly on the following parameters: precipitation rate, hotplate temperature, evaporation time, snowflake type, and density. To calculate the coincidence probability, the same data introduced above is considered with a given hotplate temperature of 104$^{\circ}$ C. With and without overlapping the time series of the area and an average temperature of hydrometeors during complete evaporations is shown in Figure R1a,b below. When compared to a typical evaporation cycle for a single frozen hydrometeor, overlapping is indicated by a significant decrease in temperature and increase in area within a normal cycle of evaporation. By applying these conditions, the probability of coincidence is calculated. A second method takes into account the size distribution, which provides a vertical structure of hydrometeors based on precipitation rate. An overlap is counted if the evaporation time of any hydrometeors is greater than the average time between two

consecutive hydrometeor in the vertical direction. Using these two methods, negligible overlaps were observed for a precipitation rate of ~1 mm hr⁻¹, and a maximum of 4.9 % coincidence probability was observed during the highest SWE rate 15.6 mm hr⁻¹ and it is given in Figure R2. Note that during instances of overlap, in contrast with optical distrometers, the DEID does not lose measurement of the primary quantity of hydrometeor amount, in this case mass. The DEID provides a combined mass from Eq. 17. Total mass estimation is unaffected although individual particle calculations such as mass, size, and density are.

For data where overlap is identified, these measurements are not considered in the probability and size distributions etc.

[Figure]

Figure R1. Example time series of individual hydrometeor area and average temperature during complete evaporation for a case (a) without overlap and a case (b) with overlap.

[Figure]

Figure R2. Probability of coincidence as a function of SWE rate.

**Reviewer Comment 2:** *One of the most critical parameters in all of the equations to predict density and mass, is the time to completely evaporate a crystal; and yet only a single figure (Fig. 5) shows this parameter for a single water droplet. I would like to see some actual Size vs time for ice crystals in field experiments so as to illustrates the variability with size, mass and density. These times also help determine the frame rates and probability of coincidence, so a lot more needs to be discussed about their importance for deriving the parameters that are being advertised as available from this instrument.*

**Response:** Evaporation time, defined as the time taken for complete evaporation of an individual hydrometeor, depends on the following parameters: the temperature of the hotplate, the roughness of the hotplate, ambient conditions including wind velocity, temperature, and humidity, etc. 45718 snowflakes were considered for plots of evaporation time vs diameter, mass, and density as shown in Figure R3 a,b,c. The plate temperature was set to $104^\circ$ C and the thermal camera sampled at a frame rate of 12 Hz. The median values with lower and upper quartiles for the evaporation time is 2.41 [1.25, 5] sec. Hence, this range of time scale minimizes uncertainty in measurement for all type of hydrometeors at given hotplate temperature and frame rate.

[Figure]

Figure R3. (a) mass, (b) density, and (c) equivalent diameter as a function of evaporation time of water droplets originating from melting snowflakes.

**Reviewer Comment 3:** *The camera frame rates that are mentioned vary quite a bit, from 5-240. It appears that the higher frame rates were used just to validate certain aspects about detection and melting rates, but operationally much lower rates are used. Why? This raises a very important issue that is not addressed: "What is the processing time?". With 1.2 Mpixels to process from each frame, how long does it take to identify and accept/reject each particle in a frame, what are the filtering criiteria and has fast can all the derived parameters be output? Is this near-realtime or does this require substantial pot-processing time so that the applications can only be for research and not for operational applications?*

**Response:** Higher frame rates were used to validate aspects of particle detection and the melting rates etc. and a lower rate (12 Hz) was used in field observations. To determine a thermal camera frame rate that would capture the widest possible range of hydrometeor types , an experiment was performed during a snow event at Red-Butte Canyon on 25 March 2020. The thermal camera was operated at a frequency of

60 Hz with the plate temperature set to 104° C. The total mass of hydrometeors was estimated using two different algorithms, from the total mass in each frame, a summation of the mass of each particle. The difference in total mass between the two algorithms can arise due to rejection of hydrometeors with an evaporation time less than three consecutive frames or from incomplete evaporation at the end of sample period. A period with a length of three frames (0.25 sec) was selected as a minimum for performing an accurate mass measurement. The total mass of hydrometeors that fell on the hotplate within half an hour was calculated using sampling frequencies of 1, 2, 3, 6, 10, 12, 15, 20, 30, and 60 Hz and shown in Figure R4. Using the frame-by-frame method the calculated total mass at 12 Hz frequency is 99.82 % of the total mass calculated at 60 Hz, so it is this method that is used for SWE accumulation calculations. Using the particle-by-particle method the calculated total mass at 12 Hz frequency is 94.79 % of the total mass calculated at 60 Hz.

[Figure]

Figure R4. Normalized total mass that is total mass at different frequency divided by total mass at highest frequency is plotted against sample frequency.

Sampling at 60 Hz could also be done, but it is less practical operationally. For a ~ 1.2 Mpixels camera resolution, the processing time for each frame is approximately 0.015 sec. The average size of the data for a one-hour period is 1.3 Gb and the associated processing time is approximately 11 minutes. Selecting a frame rate of 12 Hz, in part, assures that the DEID can operate as a real-time instrument. Hence, the 12 Hz represents a cost benefit balance between accuracy of the measurement and time and storage costs.

**Reviewer Comment 4:** *How do you avoid measuring snow lifted from nearby surfaces, i.e. how do you know that you are measuring freefalling snowflakes?*

**Response:** At the Alta Collins site location, the DEID is collocated alongside instrumentation deployed at the long-running Collins Snow Study Plot (CLN), which is a well-protected snow study site located at the upper terminus of Little Cottonwood Canyon, averaging 1300 cm of snowfall annually and 17.4 days with at least 25 cm of snow per winter. The full record from CLN spans 41 years (January 1980–April 2021),

and the last 21 seasons include a complete record of automated hourly precipitation observations (Alcott and Steenburgh 2010).

This site was chosen in part to avoid the additional measurement of windblown snow that would typically be lifted from exposed terrain features. However, we did not do anything to specifically avoid measuring lifted snow other than using this well-sheltered area along with keeping the plate surface elevated ~1.25 m above the ground surface. Blowing snow is likely to have a distinct signature by way of particle clustering and size. In the current state, no distinction has been made between the characteristics of freefalling and lifted snow. If there is a flux of precipitation falling downwards onto the plate it will be measured whatever its origins.

**Reviewer Comment 5:** *Can you measure graupel or snow pellets that bounce?*

**Response:** This is an excellent question. Snow particles bouncing from the heated plate are a function of the following two-time scales: (a) the contact time between plate and snow particle and (b) the melting time of the initial contacting layer of the snow particle. There is a competition between the contact time and melting time. Contact time decreases with increasing density of a snow particle, and melting time increases with increasing density of snow particles. For a given density of snow particle (74 kg m$^{-3}$), the contact time is O ($10^{-1}$ sec), and the melting time of a 100 μm thick layer is O ($10^{-3}$ sec). When a snow particle melts, the normal reaction force of the surface to the snow particle is weakened. A roughened plate surface and the surface tension between plate and initial melted water layer of the snow particle helps to hold the snow particle in place after impacting the heated plate.

From experimental observations between November 2019 and April 2021, there were no observed incidents of bouncing from the heated plate. The maximum observed density of snow particles was estimated to be 632 kg m$^{-3}$. There is the possibility for bouncing for higher particle densities, plausibly hail, but these were not observed. As another point of evidence the total SWE accumulation was compared with manual measurements from the Alta-Collins snow-study plot. A windshield was implemented around the manual bucket to increase catchment efficiency. The correlation between DEID and the manual SWE measurement is 0.997 for 10 snow events.

**Reviewer Comment 6:** *Snowflakes form on aerosols and scavenge them, as well. These will remain as residue after the crystal melts. What is the impact on the measurements and how does this issue get addressed? How about issues of condensation on optical surface/components of the camera? Turbulent flow around the camera will likely deposit blowing snow on camera surface.*

**Response:** Indeed, due to its location east of the Great Basin, Salt Lake City and surroundings is particularly prone to dust storms. Nonetheless, based on observations from the Alta study plot and Red Butte Canyon from winter 2019 to spring 2021, aerosol residue was noted only following a couple of dust storms. Dust storms left static residue on the hotplate that was recognized by the thermal camera as a brighter signature than the usual dark metal background. To restore accurate measurement the dust residue was cleaned from the hotplate surface by
(1) Manually rubbing the plate with fresh snow and a clean cloth.

(2) Self-cleaning during snow events – the hotplate is briefly turned off remotely during the beginning of a storm and turned on after an accumulation of ~ 2 mm of fresh snow on the plate.

It is common for a very small (~ 0.001%) area of the hotplate to exhibit residue visible in the thermal imagery that remains.  Typically, these bright spots can be removed computationally. Using the frame-by-frame method, total mass due to the residues was subtracted in each frame and the total area of all residues was subtracted from the hotplate area. Using the particle-by-particle method, all hydrometeors must complete the cycle of evaporation where the area of hydrometeor must be zero at the beginning and end of the evaporation. Given residues do not evaporate, residues are not counted and the hotplate sampling area is reduced by subtracting the total area of the residues.

Condensation or accumulation on the thermal camera was reported during the entirety of observations only once.  During an extreme snowfall event the thermal camera was blocked by blowing snow for one hour and 20 minutes during a single storm that had produced ~ 216 cm after three days of snow accumulation by this point in time.

---

## Author Comment (AC2)

**Reviewer 2 comments and response**

We appreciate the reviewers insightful comments. We have tried to address them all carefully below in a point-by-point manner and believe the paper has been improved as a result.

**Review Comment 1:** Is there an operating range for the instrument? What is energy consumption as a function of ambient temperature? This would shed light on the requirements for deployments in different areas.

**Response:** The recommended operating range of the DEID for field experiments is from 104° C to 106° C for ambient temperatures ranging from -20° C to 20° C. At the Red Butte Canyon experimental site, the DEID was operated at 104° C for the entire experimental period between December 2019 and April 2020, with ambient temperatures ranging from -12° C to 10° C. At the Alta Collins site, the DEID was operated at 106° C for the entire experimental duration between October 2020 and April 2021, with ambient temperatures that varied from -20° C to 20° C. The DEID energy consumption is estimated using the following equation

$$Convective\ power\ loss\ =\ h_c A_p\ \Delta\ T,$$

where, $h_c$ is the wind speed dependent convective heat transfer coefficient for air flow over flat aluminium plate, $A_p$ is the area of the hotplate, $\Delta\ T$ is temperature difference between hotplate and the ambient air. The rate of convective heat loss rate for a range of wind speeds and temperatures is given in figure R1.

[Figure]

Figure R1. Rate of convective energy loss for different ambient temperature and wind speed.

**Review Comment 2:** What are the impacts of wind flow around the instrument for snow events? Since this is mentioned as a drawback of other instruments, should address. It seems like provided wind tunnel tests were focused on thermal effects of wind (this is good!). This should be pointed out at L295, otherwise reader could interpret this statement too far. The statement at L350+ about minimal interference with the camera seems like a stretch, but that's coming from someone that lives in a windy environment. I agree that a flat plate implies it is better than other platforms.

**Response:** At the Alta Collins site location, the DEID was collocated alongside instrumentation deployed at the long-running Collins Snow Study Plot (CLN), which is a well-protected snow study site located at the upper terminus of Little Cottonwood Canyon, averaging 1300 cm of snowfall annually and 17.4 days with at least 25 cm of snow per winter. The full record from CLN spans 41 years (January 1980–April 2021), and the last 21 seasons include a complete record of automated hourly precipitation observations (Alcott and Steenburgh 2010).

This site was chosen in part to avoid the additional measurement of windblown snow that would typically be lifted from exposed terrain features. However, we did not do anything to specifically avoid measuring lifted snow other than using this well-sheltered area along with keeping the plate surface elevated ~1.25 m above the ground surface. Lifting the plate to this height significantly reduces wind-blown effects even in non-sheltered areas (e.g., Naaim-Bouvet et al. 2014). Blowing snow is likely to have a distinct signature by way of particle clustering and size. In the current state, no distinction has been made between the characteristics of freefalling and lifted snow. If there is a flux of precipitation falling downward onto the plate, it will be measured whatever its origin.

As another point of evidence, the total SWE accumulation was compared with manual measurements from the Alta-Collins snow-study plot. A windshield was implemented around the manual bucket to increase catchment efficiency. The correlation between the DEID and the manual SWE measurement is 0.997 for 10 snow events.
At L295, it is now stated that DEID measurements are highly insensitive to the thermal impacts of winds.

Alcott, Trevor I., and W. James Steenburgh. "Snow-to-liquid ratio variability and prediction at a high-elevation site in Utah's Wasatch Mountains." *Weather and forecasting* 25.1 (2010): 323-337.

Naaim-Bouvet, Florence, et al. "Detection of snowfall occurrence during blowing snow events using photoelectric sensors." *Cold Regions Science and Technology* 106 (2014): 11-21.

**Review Comment 3:** My biggest concern with the paper is the missed opportunity to compare DEID to other common measurements. Take visibility for example... was there a forward scattering sensor on site? Provided that the instrument is sensitive to hydrometeors > 200µm, I presume there could be bias. What about MASC data? Would be great to see PDFs of select variables between the two systems. Statements like Line 310-312 could be backed up with MASC images.

**Response:** The DEID and MASC were both deployed at the Red Butte Canyon site. Unfortunately, there was no forward-scattering sensor at the site with which a visibility calculation could be compared. For a one hour period of measurement, used for comparing MASC and DEID measurements of hydrometeor maximum dimension $D_{max}$, the median values with lower and upper quartiles from the DEID and MASC are $D_{max}$ = 2.77 [1.84 4.38] mm and $D_{max}$ = 2.90 [1.93 4.89] mm respectively. The pdf is given in figure R2. A more extensive comparison between the MASC and DEID is addressed in the paper in review (Measurement report: Mass and Density of Individual Frozen Hydrometeors K Rees, D Singh, E Pardyjak, T Garrett - Atmospheric Chemistry and Physics Discussions, 2021).

[Figure]

Figure R2. PDF of $D_{max}$. 2268 snowflakes are observed using DEID and 2093 snowflakes are observed using MASC in one hours duration on Jan 26, 2020 at Red Butte Canyon.

**Review Comment 4:** Can you explain the logic between sampling rates? Why was 12 Hz decided upon for field work? Precipitation rate is mentioned, but is this determined on the fly by input from other instruments? It's unclear how the range of 2-30Hz is related to the rates quoted in Section 2.2 that mentions tests up to 120 fps / 240 Hz.

**Response:** Higher frame rates were used to validate aspects of particle detection and the melting rates etc. and a lower rate (12 Hz) was used in field observations. To determine a thermal camera frame rate that would capture the widest possible range of hydrometeor types , an experiment was performed during a snow event at Red-Butte Canyon on 25 March 2020. The thermal camera was operated at a frequency of 60 Hz with the plate temperature set to 104° C. The total mass of hydrometeors was estimated using two different algorithms, from the total mass in each frame, a summation of the mass of each particle. The difference in total mass between the two algorithms can arise due to rejection of hydrometeors with an evaporation time less than three consecutive frames or from incomplete evaporation at the end of sample period. A period with a length of three frames (0.25 sec) was selected as a minimum for performing an accurate mass measurement. The total mass of hydrometeors that fell on the hotplate within half an hour was calculated using sampling frequencies of 1, 2, 3, 6, 10, 12, 15, 20, 30, and 60 Hz and shown in Figure R4. Using the frame-by-frame method the calculated total mass at 12 Hz frequency is 99.82 % of the total mass calculated at 60 Hz, so it is this method that is used for SWE accumulation calculations. Using the particle-by-particle method the calculated total mass at 12 Hz frequency is 94.79 % of the total mass calculated at 60 Hz. Sampling at 60 Hz could also be done, but it is less practical operationally.

[Figure]

Figure R4. Normalized total mass that is total mass at different frequency divided by total mass at highest frequency is plotted against sample frequency.

**Review Comment 5:** What are the computing requirements like? It was unclear whether the imagery is processed in real-time or not, and if so, what type of resources are needed.

**Response:** For a ~ 1.2 Mpixels camera resolution, the processing time for each frame is approximately 0.015 sec. The average size of the data for a one-hour period is 1.3 Gb and the associated processing time is approximately 11 minutes. Selecting a frame rate of 12 Hz, in part, assures that the DEID can operate as a real-time instrument. Hence, the 12 Hz represents a cost benefit balance between accuracy of the measurement and time and storage costs.

**Review Comment 6:** Figure 12: Is there any significance to the width of the heavy/light snow columns? It seems like these could be broadened.

**Response:** Figure 12 is modified.

Minor comments:

Line 25: Might be worth highlighting the challenges of other instruments for snow/wind? See associated references that could be added to this section.

Parsivel:

Battaglia, A., Rustemeier, E., Tokay, A., Blahak, U., & Simmer, C. (2010). PARSIVEL Snow Observations: A Critical Assessment. *Journal of Atmospheric and Oceanic Technology*, *27*(2), 333–344. https://doi.org/10.1175/2009JTECHA1332.1

Loeb, N. and A. Kennedy, 2021: Blowing Snow at McMurdo Station, Antarctica During the AWARE Field Campaign: Surface and Ceilometer Observations. *J. Geophys. Res. Atmos.*, 126, e2020JD033935.

MASC:

Fitch, K. E., Hang, C., Talaei, A., and Garrett, T. J., 2020: Arctic observations and numerical simulations of surface wind effects on Multi-Angle Snowflake Camera measurements, Atmos. Meas. Tech., 14, 1127–1142.

**Response**: Added

Figure 4 caption: water droplets or water and ice droplets?

**Response:** In wind tunnel, only water droplet was used.

L230: Is there an extra – before Collins?

**Response:** Corrected

L235: Please clarify- what is a sample referring to if there were 2000 snowflakes or rain drops contained within? I think this is answered at L358... just make sure this is clarified earlier on.

**Response:** For each trial, 2000 snowflakes/raindrops were collected during continuous precipitation and the sample collection time varied from about 5 to15 minutes.

---

## Author Response (AR2)

**Reviewer 2 comments and response**

We thank the reviewer for the additional comments and suggestions. Please find our responses below.

**Referee Comment 1:** What's the power needed (e.g. watts) at -20 C vs. 20C? This is what I meant to convey in my original comment.

**Response:** The rate of energy loss due to convection from a hotplate with an area of 0.0056 $m^2$ under 5 m $s^{-1}$ winds is ~13 W at an ambient temperature of $20^o$ C and ~20 W at an ambient temperature of $-20^o$ C.

**Changes in manuscript:** The rate of energy loss due to convection from a hotplate with an area of 0.0056 $m^2$ under 5 m $s^{-1}$ winds is ~ 13 W at an ambient temperature of $20^o$ C and ~ 20 W at an ambient temperature of $-20^o$ C.

**Referee Comment 2:** While you talk about wind in the sense of blowing snow (lofted from the ground), what about the potential for size/mass sorting in falling snow juxtaposed with stronger winds? You may not know, but should at least acknowledge this could exist, and is subject of future work (specify performance in wind in final paragraph).

**Response:** The DEID can potentially differentiate between snow that has been lofted from the ground and free-falling snow. On a particle-by-particle basis we observe variations in the density of each snowflake while the ensemble average remains relatively constant. Intuitively, the size and mass of the snow that is available for transport will be a function of the wind speed and the associated force available for lofting and redistribution of snowflakes with differing masses and sizes. While we currently do not differentiate between freshly deposited snow and snow from wind loading, we suspect that depending on wind speed, lofted snow may have more or less variation in size and mass than what we see in free-falling snow. Distinguishing the signatures of blowing snow and free-falling snow is a topic of future work.

**Changes in manuscript:** Distinguishing the signatures of blowing snow and free-falling snow is a topic of future work.

---

## Author Response (AR3)

We have fixed the references as requested by the editor.